behaviour/bioengineering

real-time data analysis, video tracking, ants, animal behaviour

**Author for correspondence:**
Serafino Teseo
e-mail: steseo@ntu.edu.sg

# Integrating real-time data analysis into automatic tracking of social insects

Alessio Sclocco[1,2], Shirlyn Jia Yun Ong[1], Sai Yan Pyay Aung[1] and Serafino Teseo[1]

[1]School of Biological Sciences, Nanyang Technological University, Singapore
[2]Netherlands eScience Center, Amsterdam, North Holland, The Netherlands

AS, 0000-0003-3278-0518; ST, 0000-0002-0797-1032

Automatic video tracking has become a standard tool for investigating the social behaviour of insects. The recent integration of computer vision in tracking technologies will probably lead to fully automated behavioural pattern classification within the next few years. However, many current systems rely on offline data analysis and use computationally expensive techniques to track pre-recorded videos. To address this gap, we developed BACH (Behaviour Analysis maCHine), a software that performs video tracking of insect groups in real time. BACH uses object recognition via convolutional neural networks and identifies individually tagged insects via an existing matrix code recognition algorithm. We compared the tracking performances of BACH and a human observer (HO) across a series of short videos of ants moving in a two-dimensional arena. We found that BACH detected ant shapes only slightly worse than the HO. However, its matrix code-mediated identification of individual ants only attained human-comparable levels when ants moved relatively slowly, and fell when ants walked relatively fast. This happened because BACH had a relatively low efficiency in detecting matrix codes in blurry images of ants walking at high speeds. BACH needs to undergo hardware and software adjustments to overcome its present limits. Nevertheless, our study emphasizes the possibility of, and the need for, further integrating real-time data analysis into the study of animal behaviour. This will accelerate data generation, visualization and sharing, opening possibilities for conducting fully remote collaborative experiments.

## 1. Introduction

Scientists apply automated pattern analysis to images of experimental animals to extrapolate measures of their behaviour [1–3]. Annotating behaviours using computer algorithms, rather

than employing human labour, increases experimental reliability because the performance of machines varies less than that of humans. In addition, machines easily manage to focus simultaneously on multiple individuals for long periods of time, without suffering from tiredness. This allows generating otherwise unachievable data amounts in short time windows.

In the last decade, automatic video tracking systems have emerged as ideal tools to investigate the behaviour of ants [4–10]. These live in compact societies that thrive in the laboratory and easily adapt to experimental set-ups. Automatic trackers can, therefore, scrape a significant amount of information just by scanning images of their colonies. Researchers usually employ tracking systems to analyse groups of ants moving in two-dimensional arenas, tagging each individual with unique identity markers (e.g. QR or ArUco matrix codes [7,8,11,12] or combinations of painted colour dots [4–6,13]) and using cameras to take images from top and/or bottom [14]. Many video trackers rely on inferring the position and orientation of insects via the detection of individual tags in images taken at given time intervals [7,12]. From tag positions, experimenters can then estimate a variety of informative individual attributes, such as the distance travelled between data points or the general activity levels of individuals. Combined together, tag positions, relative code orientations and distances allow reconstructing social interaction networks within groups [3,15,16]. More recent 'next generation' video tracking systems integrate insect identification with analyses of their shapes and movements [13]. This not only allows to automatically identify individual animals, but also to classify some of their behaviours [13,16–18].

Although the technology underlying insect video tracking advances rapidly and significantly, many recently developed systems rely on offline data processing [11,17–21]. This means that researchers often perform tracking on previously recorded videos and analyse data *a posteriori*. This approach allows correcting errors and employing sophisticated and costly computational strategies, or even processing earlier datasets using future knowledge. On the other hand, offline data processing does not allow for simultaneous insights into already running experiments [22]. Graphics processing units (GPUs), which allow accelerating computation, begin to address this gap via integrating real-time data analysis [23,24] in the traditional way of conducting tracking-based studies on insect behaviour [12,25]. GPU-based real-time data analysis systems allow tracking experimental individuals and simultaneously analyse data while storing these for further processing, making results immediately available. Combining real-time processing with traditional experimental methods, human observers (HOs) could identify individual behavioural attributes, characterize social interactions and even determine group-level emerging properties before the end of an experiment. This not only increases the potential discovery rate by accelerating data generation and sharing, but also allows for iterative adjustments of experimental parameters based on real-time results.

With the aim of advancing in such direction, we developed BACH (Behaviour Analysis maCHine), a real-time video tracking system based on computer vision. BACH integrates existing open-source convolutional neural networks [26,27], an object identification model [26] and a system detecting matrix codes (ArUco [28,29]). We conducted a series of experiments to compare the performances of BACH and a HO, aiming to: (i) identify whether and how BACH made mistakes or omissions, and develop strategies to limit these; (ii) determine details about the subtasks BACH and HO performed, i.e. shape-based detection and code-based identification; and (iii) generate data of general relevance about the performances of humans and machines in following trajectories of individually identified insects.

Besides developing and testing another tool to conduct research on insect social interactions, this study aims to encourage researchers to integrate real-time data analysis in their systems for investigating animal behaviour. This may help achieve faster result generation and conduct fully remote cooperative research projects.

# 2. Material and methods

## 2.1. Experimental settings

We built a casing prototype using 5 mm thick black acrylic, aiming to host a single experimental ant colony per experiment (figure 1*a*). This included a nest area ($17 \times 12 \times 17$ cm) and a foraging area ($17 \times 12 \times 18$ cm, figure 1*b*) connected by a 5 mm diameter tunnel. As this study aimed to assess BACH's efficiency, and not to test biological hypotheses, we decided to focus on a single area per experimental session. Therefore, we kept the tunnel closed to prevent ants from walking between the foraging and nest areas. The casing stood on a plaster of Paris base (figure 1*c*), which we humidified

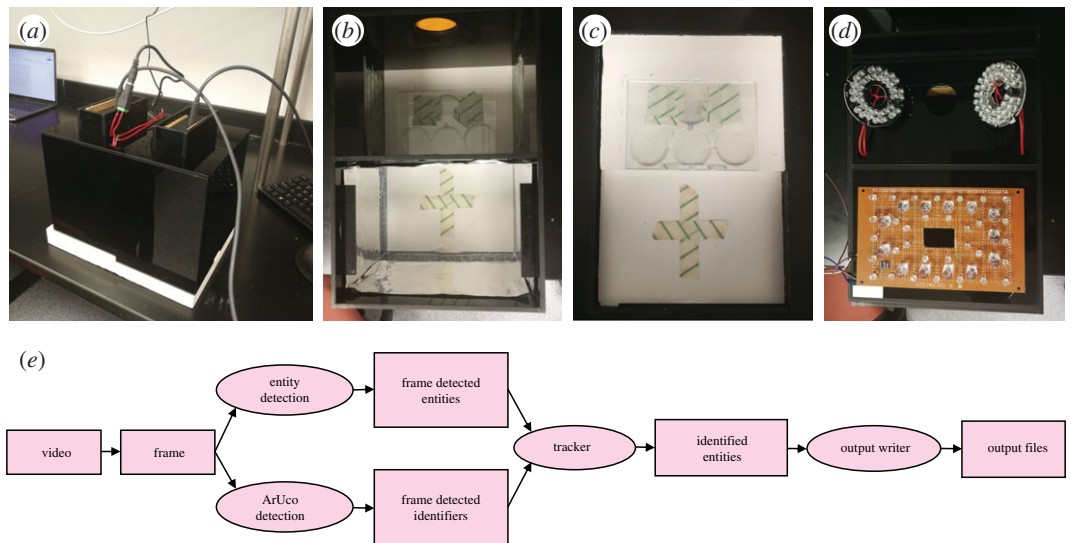

**Figure 1.** (*a*) The video tracking prototype. (*b*) The tracking set-up interior. (*c*) The plaster base with the nest area on top and the foraging area on the bottom. (*d*) The illumination system: ready-made IR LED diode panel on top, veroboard with white, IR and UV LED circuits on the bottom. (*e*) A flux diagram illustrating the functioning of BACH.

with water prior to experiments. In the nest area, we moulded three interconnected circular chambers in the plaster base, covering these with a glass lid. We painted the facing-down side of the lid with Sigmacote (Sigma) to prevent ants from climbing onto the ceiling and walking upside down. In the foraging area, ants walked on the plaster surface, and Fluon (Sigma) on the chamber walls prevented them from escaping. We positioned webcams (Logitech Brio) with the infrared filter removed in compartments above each area, and connected them using USB cables to a desktop computer (DELL Precision 7820 Tower Workstation upgraded with Dual GTX1080 GPUs and 128 Gb RAM). We affixed strips of white adhesive tape to the chamber walls to minimize the proportion of dark pixels in the images, which facilitated automatic exposition and focus. Two crossed strips of tape on the plaster floor of the foraging area further facilitated the camera focus.

## 2.2. Lighting

In the nest area, two ready-made circular mini panels of 36 LED diodes, oriented at 45° relative to the ceiling of the box, conveyed 850 nm near-infrared light (figure 1*d*). Invisible to ants, near-infrared light simulated the darkness of a real ant nest interior, at the same time allowing image recording via IR-sensitive webcams. In the foraging area, we employed white, UV and infrared light, conveyed via three independent circuits of LED diodes (24, 4 and 13 units for white, UV and IR light, respectively) manually soldered to a stripboard (figure 1*d*, electronic supplementary material, video S1). We set-up the lighting system to alternate UV, white and IR light in the foraging area via an Arduino microcontroller, through customizable commands.

## 2.3. Ant collection and preparation

For experiments, we collected and kept in the laboratory a fragment of a carpenter ant colony (genus *Camponotus*). We printed, hand-cut and glued 20 $2 \times 2$ mm $4 \times 4$-pixel ArUco markers [28–30] (from 0 to 19, obtained via http://chev.me/arucogen/) on the abdomens of ice anaesthetized ant workers ($6.81 \pm 0.93$ mm body length, measured on 50 clearly visible individuals in three frames from different videos), using Araldite Rapid 5 min epoxy. Code side/ant length ratios varied between around 0.23 and 0.40. Researchers often mark ants on the thorax because sometimes these bend their abdomen underneath the body, hiding tags from the camera view. However, after conducting preliminary tests on thorax-tagged ants, we opted for abdomen tagging as it prevented ants from removing codes with their legs, and legs from becoming irreparably entangled in the glue. After applying tags, we immediately placed ants in an open container for recovery; finally, we either placed them in the nest chambers after a second brief ice anaesthesia, or released them from the container into the foraging

area through a Fluon-coated funnel placed in the webcam window above. We waited for ants to recover normal activity levels before recording videos.

## 2.4. Image annotation and model training

For model training and testing, we established two sets of images including ArUco-tagged and non-tagged ants, one for the nest area and the other one for the foraging area. We decided to include images of non-tagged ants in order to track them even in case they lost their codes, or in case their codes became invisible to the camera for some reason. In addition, this prevented the algorithm from relying on codes themselves, and for example, learning to identify some specific codes and not others. We extrapolated images from videos taken in settings identical or similar to those of the experimental tracking. For training and validation, we used, respectively, 1551 and 53 images in the nest, and 2597 and 108 in the foraging area. We manually annotated ant images using the software Yolo_mark [26], selecting rectangular ant-including areas.

## 2.5. Video recording

We randomly selected four sets of around 20 workers of various sizes (one set for each of four recording days). This allowed us to account for random variation stemming from individual behaviour and from the manual positioning of the matrix markers on ant bodies. We recorded videos at 10 frames s$^{-1}$ for 25 s and with a resolution of 4096 × 2160 pixels, in three different conditions: (i) foraging area exposed to infrared light; (ii) foraging area exposed to visible plus UV light; and (iii) nest area exposed to infrared light. For each of the three conditions, we recorded 10 different videos for each of 4 days (120 videos in total), with 10 min pauses in between. We have uploaded three examples of videos, one for each condition, at http://doi.org/10.5281/zenodo.4541413.

## 2.6. Tracking

Using the two models resulting from the training (nest: https://zenodo.org/record/4525517; foraging area: https://zenodo.org/record/4525643), BACH (https://github.com/isazi/bach, code with instructions in the README file) tracked ants in two consecutive steps (figure 1e). First, it tried to detect each ant using the YOLO deep neural network model [26,27], processing each frame of the input video independently but in the same order as frames appeared in the video. Then, it added to a list all the detected entities for which the probability of looking like an ant exceeded a user-defined threshold; for each detected entity, the information stored in the list included the coordinates of the top-left and bottom-right vertices of a rectangular bounding box drawn around each ant. In the second step, BACH processed the same frame using OpenCV [31] to detect ArUco markers. Finally, it stored each detected marker in a list containing marker IDs and the $xy$ coordinates of their central point (the intersection of the diagonals of the bounding box).

BACH then processed the list containing the detected entities and the one containing the detected ArUco markers. First, it compared the newly detected entities to the list of those detected in the previous frames, to measure potential overlaps between their bounding boxes. If the bounding box of a newly detected entity did not overlap with any already known box, BACH added it to the system as a new ant with ID '−1'. If the bounding box overlapped with one or more boxes with an already known identity, BACH did not add it to the system as a newly detected entity, updating instead the position of the ant with the largest overlap with the new detection. In addition, it also marked the known ant as 'seen' for the current frame. After BACH processed all entities, it engaged in a similar process for the detected ArUco markers, assigning to an entity each detected marker whose central point fell inside its bounding box. In case the central point of a marker fell in the intersection of two or more bounding boxes, BACH assigned it to the entity whose central point lay the closest to the marker's central point. Each entity kept track, using an ordered list, of the count of all ArUco markers assigned to it, and used the ArUco marker with the highest count as its own ID. This ensured that the omission or incorrect detection of a marker did not affect the status of a previously detected and identified ant, allowing BACH to attribute identities to ants even when it could not see their matrix codes.

BACH deleted entities still present in the system if it could not detect them for a user-defined number of frames. In other words, in the event that a given ant detected in a frame became undetected in subsequent frames, BACH would retain its latest detection throughout a certain number of frames. We defined this as 'ghost threshold' because, in tracked videos, the bounding box of an ant becoming undetected remained empty in the location of its last detection. If BACH detected the ant again before reaching the ghost

threshold, it kept tracking it, whereas it deleted the retained detection if it did not succeed in detecting the ant again. We set the ghost threshold to one frame (0.10 s) for the tracking in the foraging area, and five frames (0.50 s) for that in the nest area. We used such values because ants in the foraging area tended to move faster than in the nest, resulting in more missed detections. Finally, BACH appended all entities still present in the system to the output file of the tracker, which recorded the position of each identified entity for each frame.

# 3. Measuring system efficiency

## 3.1. Frame extraction and ant identification by HO

We randomly selected four videos per condition taken across at least three different days. Each video consisted of 250 frames labelled 0–249. As manual ant identification required a significant amount of repetitive work, we extracted and analysed only one in ten frames (19 frames from 60 to 240). We took frame 60 as a starting point because, after activation, the camera focus stabilized by frame 60 across all videos. HO identified ants by reading their ArUco codes and using ImageJ [32] to compare their coordinates with those attributed by BACH (and not knowing the identities BACH gave to ants). This allowed HO to align ants with the coordinates provided by BACH and to manually annotate coordinates of ants that BACH did not detect. HO also classified each ant in terms of visibility of its ArUco marker (horizontal, tilted, invisible, electronic supplementary material, figure S1a–c) and blurriness (sharp, blurry, very blurry, electronic supplementary material, figure S1d–f). HO also considered codes as invisible under other circumstances, including: instances in which one or more ants partially covered the body of another ant, altering its shape and/or obstructing its ArUco code (electronic supplementary material, figure S1g); instances in which ants partially or totally climbed the foraging area walls (electronic supplementary material, figure S1h), and as they did not contrast against the black acrylic walls of the tracking set-up, BACH could not detect them and identify their matrix code; instances in which ants bended their abdomen forward in order to groom themselves (electronic supplementary material, figure S1i), which altered their shape and made their ArUco codes invisible to the cameras; instances in which ants climbed on the nest glass lid ceiling and walked upside down, which hid matrix codes under their bodies (electronic supplementary material, figure S1j).

## 3.2. Verification of HO identification and manual tracking

To evaluate and compare the performances of HO and BACH, we needed a completely accurate ant identification that we could use as a reference. Therefore, we further manually tracked all ants across all frames of the 12 analysed videos. This procedure differed from the aforementioned tracking of HO, which only analysed one frame per time without referring to other frames within the same video, and did not know the identities attributed by BACH. Contrarily, in this procedure we verified and corrected all detections, and for instances in which HO or BACH did not identify codes, we replayed videos and followed individuals until their code appeared identifiable in a preceding or following frame. In addition, we manually tracked all 'unknown' individuals that HO or BACH could never identify in any of the frames. Besides spotting misidentification errors, this further tracking allowed us to fill gaps between frames, for example when HO or BACH identified an individual at frame 60, did not identify it at frames 70, 80, 90 and identified it again at frame 100. Importantly, it also allowed retrieving the coordinates of all individuals across all frames, which in turn enabled us to classify attributes of the instances in which BACH and HO detected/identified them, or failed to do so.

Finally, during this manual verification, we discovered two peculiar cases of detection by BACH: in the first, which we defined as 'double' (electronic supplementary material, figure S1 k), BACH detected two ants as a single one; in the second, that we defined as 'double plus one', it first detected two ants as a single one, and then detected correctly one of the ants while still detecting both ants as a single one (electronic supplementary material, figure S1l). In our manual tracking, we treated double detections as equivalent to detecting only one of the ants, and 'double plus one detection' as equivalent to detecting both ants.

## 3.3. Ant walking speed calculations

To have an idea of the walking speed of each experimental individual, we used as a proxy the distance between the $xy$ coordinates of the same ant in two consecutive frames. This corresponded to the average

walking speed of the individual during the second preceding each analysed frame. As a consequence of this, we could not calculate the ant speed at the first analysed frame of each video (frame 60). Therefore, we removed frame 60 in all analyses taking into account the ant walking speed.

BACH and HO retrieved the $xy$ coordinates of ants by annotating the central point of their shape within the picture. HO did this by eye with the support of ImageJ for reading individuals' coordinates; for BACH, individuals' coordinates corresponded to the centre of a bounding box including each ant shape in a given frame. Therefore, if individual positions varied slightly across two consecutive frames (e.g. an ant moved a leg without changing location), BACH's bounding boxes changed in order to fit the new shape. This ultimately resulted in a shift of the coordinates of the bounding box's centre, which created a small, artefactual displacement the ant did not actually make. To avoid including this artefactual variation, we visually analysed a subsample of ant displacements with the corresponding speed measures, and decided to round to zero all speed values lower than $0.75$ mm s$^{-1}$.

# 4. Statistical analyses

For statistics on the number of ant detections, we used R [33] to conduct Pearson's $\chi^2$-tests. We proceeded this way, which implied considering each video frame as independent from the previous one, because BACH detected ants within each frame independently from their position in the previous frame.

For analysing identification data, we needed to consider interactive factors and/or used random intercepts that allowed fitting repeated measure designs on binomial or count data. Therefore, we used the R package lme4 [34] to implement generalized linear mixed models (GLMMs). For most models, we included individual identity nested in frame number nested in video as a random factor (in the R package lme4: 1 | video/frame/id). We used this structure because BACH identified individuals also based on their position and identity in the previous frames, while videos involved different sets of individuals.

To test the significance of interactive effects in GLMMs, we compared models including both the interaction and the single factors with corresponding models only including single factors; similarly, to test the significance of single factors in models without interactions, we compared the model including the factor with the corresponding model only including the intercept. For such comparisons, we used the R function 'anova' with 'Chisq' test specification. To test specific *post hoc* contrasts, we used the R package emmeans [35] (with Tukey HSD test and automatically adjusted $p$-values) when the factors of interest had more than two levels. When factors had only two levels, we instead referred to the contrasts appearing in the output of the R function 'summary' applied to each model. If the residual deviance considerably exceeded the residual degrees of freedom, we corrected GLMs and GLMMs for overdispersion by including each observation as an additional random factor. If models failed to converge or resulted in singular fits, we used optimizers (from the optimx R package [36,37]). We provide scripts for statistical tests in the electronic supplementary material.

# 5. Results

## 5.1. Ant detection

HO detected ants significantly better than BACH, overall (Pearson's $\chi^2$-test, $\chi^2 = 725.68$, d.f. = 1, $p < 0.001$) and in each of the three tracking conditions (Pearson's $\chi^2$-test, nest: $\chi^2 = 212.17$, d.f. = 1, $p < 0.001$; foraging area under visible light: $\chi^2 = 227.28$, d.f. = 1, $p < 0.001$; foraging area under infrared light: $\chi^2 = 288.64$, d.f. = 1, $p < 0.001$, figure 2$a$). Considering all conditions together, HO detected ant shapes in all instances, while BACH did so 83.8% of times. As HO never failed to detect ants, its accuracy did not vary across conditions. On the other hand, BACH detected 86.6% of ant shapes within the nest, 84.7% in the foraging area with visible light and 79.3% in the foraging area with infrared light, with a significant condition-specific variation in detection efficiency (Pearson's $\chi^2$-test, $\chi^2 = 27.94$, d.f. = 2, $p < 0.001$, figure 2$a$).

The walking speed of ants varied significantly across conditions (GLMM, $\chi^2 = 12.69$, d.f. = 2, $p < 0.01$, figure 2$b$). Ants walked the fastest in the foraging area under visible light ($6.71 \pm 9.43$ mm s$^{-1}$), significantly faster than the average $2.44 \pm 4.95$ mm s$^{-1}$ of ants in the foraging area under infrared light (estimate = 2.22, $z = 3.48$, $p < 0.01$) and than the average $1.19 \pm 2.20$ mm s$^{-1}$ of ants inside the nest (estimate = 2.88, $z = 4.52$, $p < 0.001$). In infrared light conditions, ant speed did not vary between nest

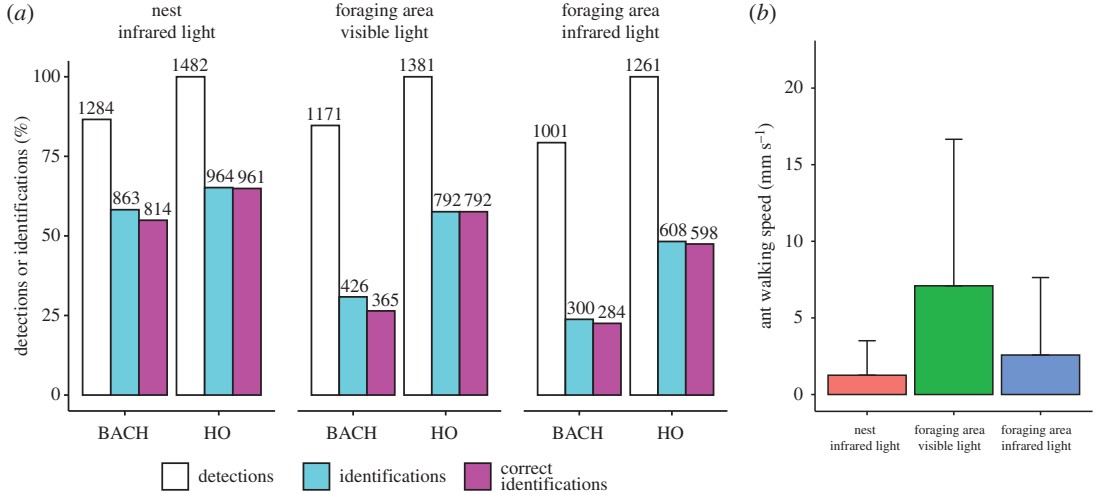

**Figure 2.** (*a*) Number of detections, identifications and correct identifications for BACH and HO across conditions. (*b*) Walking speed of ants across conditions (mean + s.d.).

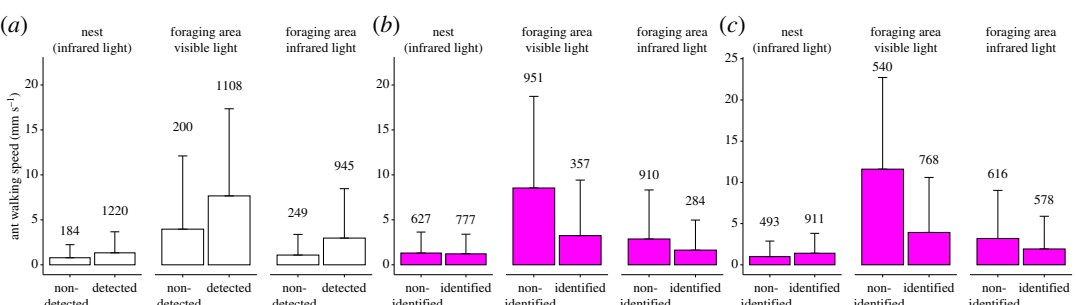

**Figure 3.** (*a*) Speed of BACH-detected and undetected ants in different conditions. (*b*) Speed of BACH-identified and non-identified ants. (*c*) Speed of HO identified and non-identified ants. Mean + s.d. across all panels.

and foraging area (GLMM, estimate = 0.66, $z$ = 1.03, $p$ = 0.55). BACH-detected ants walked faster than undetected ants, in general (GLMM, estimate = 0.07, $z$-value = 6.65, $p < 0.001$) and in each tracking condition (GLMMs; nest: estimate = 0.17, $z$-value = 3.49, $p < 0.001$; foraging area under infrared light: estimate = 0.11, $z$-value = 4.28, $p < 0.001$; foraging area under visible light: estimate = 0.05, $z$-value = 4.66, $p < 0.001$, figure 3*a*). HO detected ants in all instances, therefore, ant walking speed did not affect its performance.

Sometimes BACH detected two ants as a single one, at a rate that varied significantly across conditions (Pearson's $\chi^2$-test, $\chi^2$ = 11.31, d.f. = 2, $p < 0.01$). BACH produced 72 of such 'double detections' inside the nest (4.85% of total nest detections), 48 in the foraging area with visible light (3.47%) and 79 in the foraging area with infrared light (6.26%). 'Double plus one' detections occurred instead only once in the nest (0.13%), six times in the foraging area under visible light (0.36%) and nine times in the foraging area under infrared light (0.72%), also varying significantly across conditions (Pearson's $\chi^2$-test, $\chi^2$ = 7.47, d.f. = 2, $p$ = 0.023). In two of the videos taken in the nest area, we did not apply Sigmacote on the facing-down side of the glass covering the nest chambers, which resulted in ants walking on the nest glass ceiling. Of 238 instances of ants walking on the glass ceiling (31.31% of total detections in the two videos), BACH failed to detect ants 44 times (18.48%). Finally, sometimes ants partially or totally climbed the foraging area walls. In some of such cases, BACH could not detect them. This occurred 98 times in the foraging area in visible light and 78 times in the foraging area in infrared light (respectively, 7.09% and 6.18% of total detections in each condition).

## 5.2. Ant identification

HO identified ants better than BACH across all conditions (GLMM, nest: estimate = 0.88, $z$ = 8.61, $p < 0.001$; foraging area in infrared light: estimate = 1.89, $z$ = 17.16, $p < 0.001$; foraging area in visible

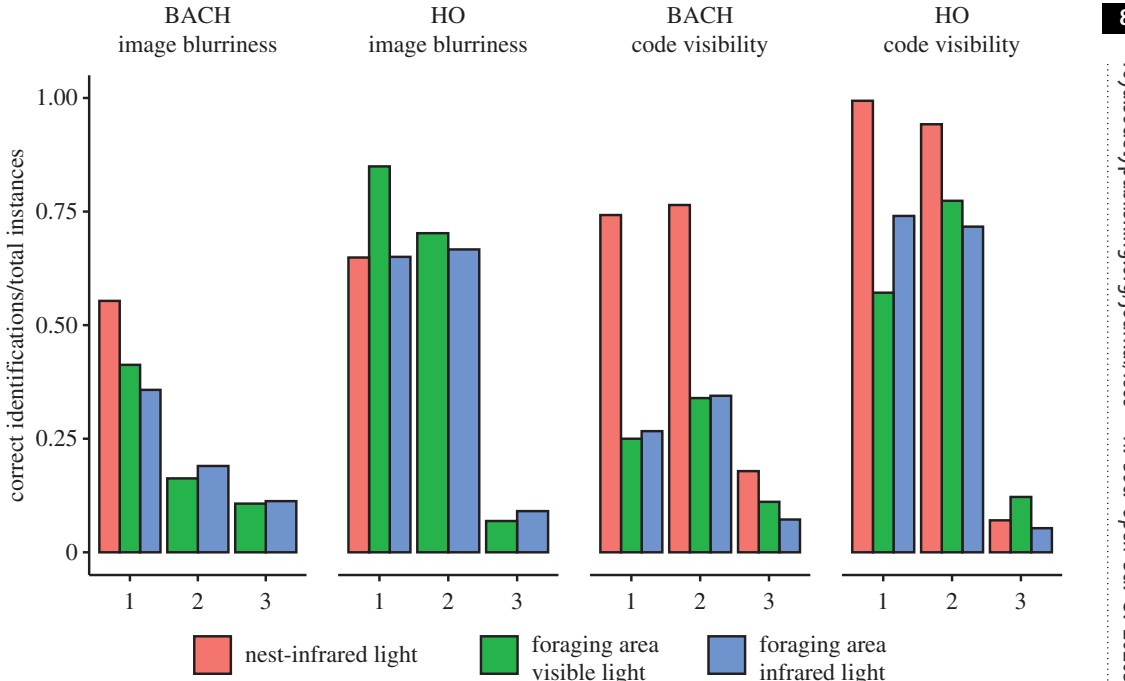

**Figure 4.** Proportion of identifications by BACH and HO across different classes of blurriness and code visibility. Numbers correspond to the different classes (sharp, blurry, very blurry for image blurriness, horizontal, tilted, invisible for code visibility).

light: estimate = 1.95, $z = 20.38$, $p < 0.001$, figure 2a). Ant walking speed, ant image blurriness and code visibility interactively affected both BACH's and HO's ant identification (ANOVAs on GLMMs, respectively: $\chi^2 = 122.81$, d.f. = 12, $p < 0.001$; $\chi^2 = 954.18$, d.f. = 12, $p < 0.001$). However, non-identified ants generally walked faster than identified ants (BACH: $\chi^2 = 139.25$, d.f. = 1, $p < 0.001$, figure 3b; HO: $\chi^2 = 237.83$, d.f. = 1, $p < 0.001$, figure 3c), tending to appear blurry in many video frames. Accordingly, ant walking speed varied significantly for different code blurriness classes (namely: 'sharp', 'blurry', 'very blurry'; ANOVA on GLMMs, 'Chisq' test, $\chi^2 = 303.61$, d.f. = 2, $p < 0.001$). In particular, sharp ants walked at lower speeds compared to blurry and very blurry ants (*post hoc* contrasts, respectively: estimate: −1.23, $z$-ratio = − 7.81, $p < 0.001$; estimate: −2.72, $z$-ratio = − 17.49, $p < 0.001$), and blurry ants walked at lower speeds compared with very blurry ants (estimate: −1.49, $z$-ratio = − 8.26; $p < 0.001$). This corresponded to different walking speeds in the foraging area (under visible light: sharp: $3.6 \pm$ 6.99 mm s$^{-1}$, slightly blurry: $6.18 \pm 8.99$ mm s$^{-1}$, very blurry: $6.57 \pm 9.09$ mm s$^{-1}$; under infrared light: sharp: $2.09 \pm 4.28$ mm s$^{-1}$, slightly blurry: $3.08 \pm 5.68$ mm s$^{-1}$, very blurry: $5.66 \pm 8.86$ mm s$^{-1}$). We only found sharp ant images in videos taken inside the nest, where ants generally walked at a lower speed ($1.19 \pm 2.20$ mm s$^{-1}$) compared with the other conditions. We concluded that BACH may have had difficulties in identifying fast-walking ants because of their blurry codes. We, therefore, dropped speed in favour of image blurriness in further analyses.

We found an interactive effect of image blurriness and code visibility on identification accuracy for both BACH and HO, overall (ANOVAs on GLMMs, respectively: $\chi^2 = 111.19$, d.f. = 4, $p < 0.001$; $\chi^2 = 1102.30$, d.f. = 4, $p < 0.001$, figure 4) and in the foraging area, both under infrared light (BACH: $\chi^2 = 71.81$, d.f. = 4, $p < 0.001$; HO: $\chi^2 = 133.98$, d.f. = 4, $p < 0.001$, figure 4) and visible light (BACH: $\chi^2 =$ 23.33, d.f. = 4, $p < 0.001$; HO: $\chi^2 = 401.59$, d.f. = 4, $p < 0.001$, figure 4). As in the nest area we only encountered sharp ant images, we only tested the effect of code visibility, which we found significant for both BACH (ANOVA on GLMMs, $\chi^2 = 460.87$, d.f. = 2, $p < 0.001$, figure 4) and HO (ANOVA on GLMMs, $\chi^2 = 1333$, d.f. = 2, $p < 0.001$, figure 4). In general, and as expected, BACH and HO tended to best identify ants in sharp images and when they could see their codes.

Both HO and BACH occasionally attributed wrong identities to ants (figure 2a). For all conditions pooled, HO identified ants in 57.46% of all instances, accurately doing so 57.15% of times. BACH identified ants in 38.53% of instances, with 35.47% of correct identifications. In the nest area, HO identifications reached 65.18% of the total, with 64.97% of correct identifications; in the foraging area under visible light, HO identified ants in 57.63% of instances, never making mistakes; finally, in the foraging area under infrared light, it identified codes 48.25% of times, with 47.46% accuracy. In the

nest area, BACH identified codes 58.23% of times, with 54.92% correct identifications; in the foraging area under visible light, it identified codes 30.84% of times, with 26.43% accuracy; in the foraging area under infrared light, identifications dropped to 23.79% of the total, and correct identifications to 22.52%. The ratio of correct to total identifications of HO significantly exceeded that of BACH across conditions (nest: Pearson's $\chi^2$-test, $\chi^2 = 47.53$, d.f. = 1, $p < 0.001$; foraging area in visible light: Pearson's $\chi^2$-test, $\chi^2 = 119.97$, d.f. = 1, $p < 0.001$; foraging area under infrared light: Pearson's $\chi^2$-test, $\chi^2 = 9.82$, d.f. = 1, $p < 0.01$). This indicated that, when identifying codes, BACH made more mistakes than HO. BACH attributed incorrect identities in 165 of 1628 identification instances (10.13%). These mistakes resulted from 27 identity exchange events between ants located in close proximity ($7.46 \pm 3.52$ mm). On such occasions, BACH did not disentangle the shapes of different individuals, for example, detecting one ant shape and part of the other ant shape as a single entity. Even if such mistaken detections lasted only for one or few frames, they ultimately resulted in identities to shift from an individual to another. BACH kept attributing the shifted identity through a varying number of frames ($6 \pm 5.15$), typically until it managed to read again the ArUco code of the identity-shifted individual. Identification errors of HO consisted instead in wrong readings of the ArUco codes.

# 6. Discussion

In this study, we developed and tested BACH, a real-time video tracking system for insects based on computer vision and matrix code recognition. We then compared the performances of BACH and a HO, evaluating their efficiency in (i) detecting ant shapes and (ii) identifying individuals via integrating the detected shapes with matrix codes. We found that, although HO employed weeks to track a fraction of the images BACH tracked in real time, it always qualitatively outperformed BACH, making significantly less mistakes and omissions. However, this varied depending on the subtask and the tracking conditions.

For ant detection only, BACH generally performed only slightly (although significantly) worse than HO across all conditions. Interestingly, BACH-detected ants walked generally faster compared with their undetected counterparts. This suggests that, with the detection models it employed, BACH achieved a better efficiency for moving rather than non-moving entities. This probably resulted from the higher shape variation of walking compared with non-moving ants, which increased the probabilities for BACH to encounter familiar shapes it already knew from the training phase. BACH also had a fairly high efficiency when detecting ants walking relatively fast, even though these appeared blurry in the frames. This means that BACH detected unfocused ant shapes nearly as efficiently as focused shapes, probably because the training image set also included blurry images of ants.

BACH had a low detection efficiency in a few cases, for example when ants partially climbed on the black walls of the set-up casing, which made them indistinguishable from the background. In future studies, we could solve this problem by using fair rather than black acrylic, and/or by including more images of ants attempting to climb the set-up's walls in our training sets. BACH's detection performance also fell when multiple ants engaged in clusters ('double' and 'double plus one' detections), because telling apart overlapping ant shapes became difficult. A recently developed tracking system (AnTrax [13,32]) has elegantly worked around this issue via retaining the identities of individuals when they join a cluster and become indistinguishable from other ants. When such individuals leave the cluster, AnTrax retrieves their identity. This process allows maintaining a coherent flow of information even without knowing individual positions within the cluster at all times. Although BACH does not have a specific cluster-solving function, it tries to identify individuals at all times, regardless of whether they form clusters or not. This function relies on the principle that, if the camera works at high FPS settings, an individual cannot move too far from its position in the previous frame. BACH, therefore, tries to associate the identity of previous detections to current detections and, at least in principle, could tell the identity of individuals even within clusters. However, doing this with high efficiency would require high-quality images that we have not produced in the current study. In future versions of BACH, we aim to increase image quality and integrate a process using individual histories to retain ant identities while they engage in clusters. BACH runs different units of code implementing its different features in parallel and independently, running them via either separate processes or threads. In this way, modules with higher computational complexity do not block modules with lower computational complexity but stricter real-time requirements, such as the tracker. Therefore, implementing further modules to take into account trajectory histories will in principle not compromise its real-time performance.

Contrary to ant detection, BACH's identification efficiency based on matrix codes varied dramatically across tracking conditions. While BACH identified ants only slightly less effectively than HO in the nest area, its performance fell to about half of that of HO in the foraging area. This probably resulted from ants walking faster in the foraging area compared with the nest, which most likely occurred because the UV and visible light, as well as the absence of shelter, disturbed the predominantly nocturnal *Camponotus* ants. Our manual tracking and verification revealed that higher walking speed produced more instances of blurry images and therefore blurred matrix codes, which BACH often failed to identify. Accordingly, and contrary to what we found for ant detection, BACH-identified ants walked slower than non-identified ants, confirming that BACH made most identification mistakes for ants walking relatively fast. To solve this issue, we should invest future efforts in facilitating matrix code identification. The fact that even HO showed a relatively low identification accuracy points first of all to the need for upgrading our hardware settings. In future work, we aim to increase light intensity to shorten cameras' exposure times, reducing the occurrence of blurry images and therefore maximizing code readability. In addition, in our videos, the foraging area surface took around half of the frame, and the nest area only one-tenth of it, leaving a large portion unused. In the future, we could easily cut off such extra space by reducing the distance between the camera and the tracking surface, which would result in a better resolution and therefore higher chances for BACH to correctly read matrix codes. On the software side, we will increase ant identification efficiency by improving the way BACH matches new detections to the current state of the system. For example, we could match new unidentified ant detections with previously detected entities that have become unidentified, and then attribute them an identity based on numbers and identities of ants tracked in previous frames. This would probably limit not only the instances in which BACH fails to detect ants, but also those in which it mistakenly identifies matrix codes.

Regarding the general validity of our results, we must stress that our experimental conditions differ significantly from those required in biologically relevant experiments. For example, *Camponotus* colonies usually include hundreds or thousands of workers (and one or multiple queens), whereas we only tested 20-individual queenless experimental groups. In addition, we tested individuals in visible light conditions that they most likely encounter only rarely in natural conditions, which resulted in stress and high walking speeds. Conducting long-term experiments in biologically sound settings would probably trigger a significantly different set of behaviours. For example, we would expect ants to avoid foraging in visible light, and in general to walk relatively slower due to lower stress levels; we would also expect higher levels of clustering due to higher densities of individuals within the nest. Therefore, ants walking at a lower average speed would in principle improve BACH's identification efficiency, but their increased tendency to cluster would probably increase multiple detections. Further tests with larger colonies and a realistic day/night alternation in the foraging area will tell how BACH performs its tracking in more natural conditions.

Similar to BACH, other recent tracking systems for insects integrate convolutional neural networks and the identification of individual tags. However, such systems differ from BACH in their functioning and scopes. For example, AnTrax [13] extracts image portions including ants from pre-recorded videos, reconstructing ant trajectories and identifying them via linking them across frames. Contrarily, BACH processes video frames in real time, detecting individual shapes and their IDs, only keeping track of their position. While AnTrax bases its accuracy on more complex and computationally heavier techniques currently less suitable for real-time processing, BACH works in real time but extrapolates less information and makes more mistakes. In addition, AnTrax can work in concert with JAABA [16] to automatically classify ant behaviours, whereas BACH does not include similar features at its current state. In the future, however, we aim to combine the Python interfaces of DeepLabCut [3] or DeepPoseKit [15] with Yolo, integrating BACH with pose and behaviour estimation. Finally, AnTrax identifies ants via colour combinations, which provides advantages but also disadvantages compared with the matrix code system supported by BACH. Only insects with sufficiently large bodies can bear readable matrix codes, and yet these restrict their movements, probably affecting their behaviour; sometimes matrix codes detach from the insect bodies, disrupting the information flow within experiments [13]. On the other hand, colour-based identification relies on experiments conducted in visible light. This does not affect virtually blind clonal raider ants, on which AnTrax specializes, but does not suit experimental work on non-blind, lucifugous or nocturnal ant species living in dark nests. As shown in our experiments on *Camponotus*, tracking light-sensitive insects in white and UV light may alter their natural behaviour.

Other automatic trackers used to study insect societies integrate computer vision and/or matrix code identification to different extents, working mainly offline. For example, a recent tracker for honeybee

colonies [17,18,20] not only uses matrix codes for individual identification, but also allows automatically and simultaneously detecting multiple mouth-to-mouth food exchanges (trophallaxis), simultaneously tracking high numbers of individuals (greater than 1000). The system, which similar to BACH integrates convolutional neural networks and matrix code identification, classifies these episodes based on the relative position and orientation of matrix codes, confirming such identifications via custom computer vision algorithms. However, it does not remember the identities of individuals for which matrix codes become invisible, but compensates the lack of this feature via confining experimental honeybees in a relatively narrow space. This prevents bees from turning or forming multi-layer clusters where individuals in upper layers cover the markers of individuals in lower layers, minimizing non-detection instances.

Another recent tracker for honeybees identifies individual custom-made tags encoding the identity of bees, also retrieving information about the bee body orientation via the tag inclination [25,38]. The system combines consecutive individual detections into short reliable tracklets and then connects them over longer gaps, using machine learning at both steps. This allows tracking back individual trajectories that also include frames with undetectable tags. While the tag recognition portion of this tracker can in principle run in real time through the use of GPUs [25], the authors still used multiple nodes of a large cluster to track honeybees offline in their experiments.

Finally, AntVis [19] employs computer vision to visualize fluxes of ants moving in the same direction, without implementing individual ant identification over extended periods of time. Conceived for observation in natural conditions, AntVis identifies individual ants by accurately tracking their trajectories within the video frame, without relying on individual matrix code tags [39]. This means that it identifies individuals as long as they continuously appear within the video frame, considering them as new whenever they exit and re-enter the frame.

While our work on BACH and other studies have provided self-evaluations of the respective systems' efficiency, several interacting factors complicate quantitative comparisons among different trackers. First, systems largely differ due to the diversity of their hardware settings and tracking conditions, which contributes to their efficiency. For example, several studies rely on high-definition industrial cameras [17,18,25,38], whereas here we only employed standard webcams modified to become IR-light sensitive. These produced relatively low-quality images and low FPS videos. As a result, while the original standalone ArUco code detector has in principle a very high efficiency [28], it failed to reach such standards in our study. Secondly, each tracker differs from the others from a software perspective, both in function and scope. For instance, different research groups employ different matrix code identifiers, sometimes developing their own algorithms. In addition, some studies integrate the identification of codes with computer vision, while others apply convolutional neural networks to matrix code identification; different systems work online, offline or both, in some cases relying on *a posteriori* tracklet integration for disentangling individual trajectories. Finally, while some groups have conceived their systems to achieve maximal accuracy regardless of the computational effort involved, the current version of BACH targets rapid data harvesting and sharing at the expense of trajectory reconstruction accuracy and cluster solving. A systematic review evaluating and comparing systems' capabilities would, therefore, require a significant amount of non-trivial considerations. From a qualitative perspective, BACH has two main differences compared with other real-time tracking systems [12,25]. First, it remembers the identities of individuals even when their identity codes become invisible, and can successfully do so as long as it continuously detects their shape. Secondly, and most importantly, it generates real-time data through relatively light computational processes. Although BACH needs to undergo upgrading from several perspectives, we developed it with the goal of implementing real-time data analysis and ameliorating animal behaviour research that relies on video tracking. Real-time data analysis shortens the delay between observation and result generation to virtually non-existent, enabling researchers to alter experimental conditions based on current results. In addition, results become immediately available for sharing, which favours collaborative projects and accelerates the discovery rate. In a conceivable design that we aim to achieve in the near future, the current BACH main module will produce positional and behavioural data in real-time from video sources, while other modules will concurrently process the generated data and produce structured information. As an example, we will manage to observe, in real-time, the development of a colony's social graph from within BACH.

Real-time data analysis increasingly benefits cooperative research in multiple fields, including radio astronomy [24,40] and physics [41,42]. Accordingly, we conceived BACH with the goal of attracting attention on how this could facilitate animal behaviour research. For example, real-time tracking systems like BACH may help animal scientists establish and maintain collaborations during periods of

mobility restrictions. Or, in the foreseeable future, could allow conducting animal behaviour experiments entirely online, with a research group broadcasting videos of animals and other groups remotely tracking and analysing them, cooperatively and in real time.

Data accessibility. Data supporting this article have been included in the electronic supplementary material.

Authors' contributions. S.T. conceived the study, designed and built the tracking system, collected data, analysed data and drafted the manuscript; A.S. conceived the study, programmed the software and drafted the manuscript; S.J.Y.O. and S.Y.P.A. built the tracking system and contributed to the tracking model implementation; all authors contributed to and approved the final draft of the manuscript.

Competing interests. We declare we have no competing interests.

Funding. This work was supported by a Presidential Postdoctoral Fellowship (grant no. M408080000) from Nanyang Technological University (NTU) to S.T.

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
