## [Peer Review File · Royal Society Open Science]

Review History

RSOS-202033.R0 (Original submission)

Review form: Reviewer 1

Is the manuscript scientifically sound in its present form?

Yes

Are the interpretations and conclusions justified by the results?

Yes

Is the language acceptable?

Yes

Do you have any ethical concerns with this paper?

No

Have you any concerns about statistical analyses in this paper?

No

Recommendation?

Major revision is needed (please make suggestions in comments)

Comments to the Author(s)

This paper presents a new, custom-made system for automated behavioral tracking in ants. Automated behavioral tracking and machine learning for behavioral analyses is an exploding field of research, and BACH is a welcome addition to the growing list of such tools. However, I have several comments, concerns, and questions regarding the manuscript, listed below, most of which can be addressed without performing additional experiments.

Major comments:

1) The paper's main claim of novelty is based on its use of real-time image analysis, but one of the first tracking systems used in ants already implemented online image analyses (Mersch, Crespi, Keller, 2013. *Science*). The claim that "most current systems rely on offline data analysis" (and the focus on this aspect throughout the manuscript) therefore seems overstated. Please note that I don't feel novelty is essential, as several tracking systems have been developed in parallel in different labs over the last decade, and the results of those efforts are worth sharing, irrespective of any redundancy in function.

Along the same lines, the authors focus heavily on the comparison between BACH and Antrax, while largely ignoring earlier, more similar (barcode based tracking in infra-red light) systems.

Can the authors discuss other tracking systems more broadly? Beyond Mersch et al., some previous work is worth citing and discussing, including (but not limited to):

- Automated monitoring of animal behaviour with barcodes and convolutional neural networks.

Tim Gernat, Tobias Jagla, Beryl M Jones, Martin Middendorf, and Gene E Robinson. *bioRxiv* posted 27 November 2020 doi:10.1101/2020.11.27.401760.

- Boenisch, Franziska et al. (2018). "Tracking All Members of a Honey Bee Colony Over Their Lifetime Using Learned Models of Correspondence". In: *Frontiers in Robotics and AI* 5, p. 35.

2) I see the paper's main contribution as providing the first elements of a tool that might in the future be used by other groups working on collective behavior. Therefore, the authors should provide enough information to allow potential users to assess whether BACH is appropriate for their research system and questions, and if so, to easily test it. With this in mind, some important information is currently missing, hard to locate, or presented in insufficient details:

- As far as I can tell, there is little information provided on Github on how to start using the software (the README file appears to be minimal). What are the minimum requirements, supported operating systems, etc.? Will BACH run on a computing cluster? All that information should be available to potential users.

- It is crucial for potential future users to understand the relative performance of BACH compared to existing systems. In that respect, a table summarizing these differences in quantitative terms (e.g., rate of missed ants, and false positives, etc. for BACH and other tracing systems) would be useful to get a rapid, "compact" overview.

- It would be useful for the reader to get a sense of the size of the tags relative to the ants, of the size of the ants relative to the setup, and of the walking speed of the ants (as a basis of comparison with their study species); I was surprised by the lack of example video of ants moving about the setup (In understand the tracking algorithm doesn't save videos, but presumably, recording such a video separately would be reasonably straightforward). Instead of showing the setup (Fig. 1a-c) and ants (Fig. 1e) separately, could the authors show a short supplementary video of an actual experiment, or at least an image of the setup with tagged ants in it? If Fig. 1e stays in the MS, it should also have a scale bar.

- Recording: what image size was used?

Minor comments:

L. 97: Why was the tunnel kept closed? Presumably, its function is to allow movement between the nest and foraging arena and keeping it closed would prevent normal colony behavior.

L. 121: I was surprised to see the authors tagged the abdomen instead of the thorax of the ants, because the abdomen can bend underneath the body and the tag become invisible when this happens. Can the authors explain why they chose this unusual method?

L. 128: Why were non-tagged ants included in the training set? That seems like something that might decrease the performance of the algorithm.

L. 172; This section would benefit from being placed earlier in Material and methods (before the description of BACH), along with all the technical information regarding recording (frame rate, image size)

L. 436: The main advantages are not entirely convincing to me. The first advantage is also implemented

I wonder to what extent the performance of BACH was affected by the behavior of the ants. It looks like ants were stressed (recently moved to a new, sometimes brightly lit arena), and were not given much time to settle. Can the authors discuss how BACH would be expected to perform under more natural conditions (larger colonies, presence of a queen, natural foraging instead of placing ants in the foraging arena and preventing them from entering the nest)? My feeling is that some aspects might be improved (e.g. because the average walking speed might be lower) but other could be made much more challenging (e.g., clustering of ants and occlusion of tags will increase in the nest with “real” colonies).

L. 328: the surprisingly low performance of human observer makes me wonder if the performance of the tracking algorithm is not primarily constrained by poor video quality (i.e. hardware limitation) in the first place.

L. 382: I’m not sure I understand the claim that BACH’s real-time performance will not be compromised. Can the authors develop the arguments? What is meant by “module”?

L. 436: I’m not sure I understand and agree with the claim that BACH’s main advantage compared to other tracking systems is that it “remembers the identities of individuals even when their identity codes become invisible, and can successfully do so as long as it continuously detects their shape”. As far as I understand this sentence, antTrax does exactly this: it assigns IDs to shape-based trajectories, even when the individual tag is not visible in all frames in the trajectory. As noted above (see main comment), I’m also not sure about the second claim (“BACH generates data in real time through relatively light computational processes”). The light computational processes might indeed be a good argument, but in that case, I would back this claim with numbers and quantitative comparisons with other software. Generally, I would encourage the authors to put less emphasis on claims of novelty.

Review form: Reviewer 2

Is the manuscript scientifically sound in its present form?

Yes

Are the interpretations and conclusions justified by the results?

Yes

Is the language acceptable?

Yes

Do you have any ethical concerns with this paper?

No

Have you any concerns about statistical analyses in this paper?

No

Recommendation?

Major revision is needed (please make suggestions in comments)

Comments to the Author(s)

General comments:

The authors describe a newly developed real-time ant tracking system that uses a CNN to detect ants, and a combination of barcodes and a unique and lightweight tracking algorithm to identify them. They evaluate their tracking system in a pleasingly thorough and rigorous way, and highlight avenues for improvement.

My main concern relates to the novelty of their approach. Real-time social insect tracking systems already exist. For example, the very first ant tracking system by Mersch et al. (2013) processes it's videos in real time, using only barcodes. More recently, Wild et al. (2018) described a system that uses barcodes and a CNN to track honey bees in real time. I suggest the authors compare their tracking system to existing real time tracking systems and highlight how this work advances the field. This does not require new data; it could be done verbally in the Discussion.

Secondly, current social insect tracking systems are able to reliably follow >1,000 individuals. It is fine to evaluate a tracking system with just 20 individuals, as was done here, but then the manuscript should either state that the system is designed for small colonies or it should extrapolate how well critical components (i.e., the CNN for detecting the ants and the algorithm for maintaining identities when the barcode cannot be read) will perform when more individuals are being tracked.

Specific comments:

Lines 1-2: The title is slightly misleading, because BACH tracks ant identities and positions, not behavior. I suggest modifying this to "Integrating real-time data analysis into automatic social insect tracking".

Lines 19-20: Reading the abstract the first few times I found the phrase "concerning computer vision-based ant detection only" confusing. It only made sense to me after reading the main text. I suggest to delete this.

Lines 38-39: The discussion would benefit from a slightly broader perspective. I suggest to change "ants" to "social insects" and cite some of the honey bee tracking systems (e.g. Landgraf et al. 2015 *Frontiers in Robotics and AI* and Gernat et al. 2018 in *PNAS*).

Lines 39-41: The behavioral repertoire of social insects is commonly thought to be rich, not narrow, and their social interactions are complex, not simple. I suggest to change this sentence accordingly.

Lines 41-42: If it was easy to scrape a significant amount of information just by scanning images, automatic trackers would do a better job at this. I suggest to delete "easy".

Lines 50-53: For completeness sake, and as a pointer to the next "next generation", this should also cite Gernat et al. 2020 on bioRxiv, which uses AI to detect honey bee behavior.

Lines 54-68: The first sentence of this paragraph should cite the tracking systems that rely on offline processing. More importantly, though, the authors should mention which systems are already capable to process their footage in real-time, such as Wild et al. 2018 on arXiv and Mersch et al. 2013 in Science.

Lines 82-83: The distinction between efficiency and performance is unclear. Moreover, lines 78-79 suggest that BACH is much more efficient than HO, which seemingly contradicts this statement.

Lines 175-177: Please specify the video resolution, so BACH can be better compared to other real-time capable tracking systems, and so it is possible to approximate the size of the barcode in the videos.

Line 181: The abbreviation "HO" is defined at multiple places in the manuscript. Defining it once would be sufficient.

Line 322: I was unable to find Table S1.

Lines 393-395: A better approach would be to use brighter lights so the camera will use a lower exposure time, which will result in less motion blur. Using a higher frame rate will not help to increase the detection rate.

Lines 479-480: It is unclear what the error bars represent.

References:

Mersch, D. P., Crespi, A. & Keller, L. Tracking individuals shows spatial fidelity is a key regulator of ant social organization. *Science* 340, 1090-3 (2013).

Wild, B., Sixt, L. & Landgraf, T. Automatic localization and decoding of honeybee markers using deep convolutional neural networks. 1-20 (2018).

Decision letter (RSOS-202033.R0)

Dear Dr Teseo

The Editors assigned to your paper RSOS-202033 "Integrating real-time data analysis into automatic tracking of social insect behaviour" have now received comments from reviewers and would like you to revise the paper in accordance with the reviewer comments and any comments from the Editors. Please note this decision does not guarantee eventual acceptance.

Please submit your revised manuscript and required files (see below) no later than 21 days from today's (ie 27-Jan-2021) date. Note: the ScholarOne system will 'lock' if submission of the revision is attempted 21 or more days after the deadline. If you do not think you will be able to meet this deadline please contact the editorial office immediately.

on behalf of Dr Agustina Gómez-Laich (Associate Editor) and Kevin Padian (Subject Editor)
openscience@royalsociety.org

Associate Editor Comments to Author (Dr Agustina Gómez-Laich):

The manuscript entitled "Integrating real-time data analysis into automatic tracking of social insect behaviour" has now been seen by two reviewers. Both of them found the work interesting and commend the authors for developing a new tool for automatic video tracking. However, reviewers raised several concerns and requested some substantial changes. One of the principal concerns both reviewers have is related to the novelty of the approach. Authors emphasize that most current systems rely on offline data analysis instead of tracking insects on a real time basis and present BACH as a novelty to address this gap however, one of the first tracking systems used in ants several years ago (2013) already implemented online image analyses. Thus, both reviewers recommend authors to discuss more in detail about other systems, compare BACH to existing real time tracking systems, highlight the benefits of using this new tool and put less emphasis on claims of novelty. In addition, reviewer#1 considers that important information is currently

missing, hard to locate, or presented in insufficient details. This reviewer suggest authors to give more detailed information about how BACH works (e.g. minimum requirements, supported operating systems) and include a short supplementary video or image of the setup with tagged ants in it to allow the reader to get a sense of the size of the tags, of the setup and of the walking speed of the ants. With this information potential users would be able to evaluate if this tool is appropriate for their research system and test it. Finally, another major concern both reviewers have is how BACH would be expected to perform on larger colonies and under more natural conditions.

Reviewer comments to Author:

Reviewer: 1

Comments to the Author(s)

This paper presents a new, custom-made system for automated behavioral tracking in ants. Automated behavioral tracking and machine learning for behavioral analyses is an exploding field of research, and BACH is a welcome addition to the growing list of such tools. However, I have several comments, concerns, and questions regarding the manuscript, listed below, most of which can be addressed without performing additional experiments.

Major comments:

1) The paper's main claim of novelty is based on its use of real-time image analysis, but one of the first tracking systems used in ants already implemented online image analyses (Mersch, Crespi, Keller, 2013. Science). The claim that "most current systems rely on offline data analysis" (and the focus on this aspect throughout the manuscript) therefore seems overstated. Please note that I don't feel novelty is essential, as several tracking systems have been developed in parallel in different labs over the last decade, and the results of those efforts are worth sharing, irrespective of any redundancy in function.

Along the same lines, the authors focus heavily on the comparison between BACH and Antrax, while largely ignoring earlier, more similar (barcode based tracking in infra-red light) systems. Can the authors discuss other tracking systems more broadly? Beyond Mersch et al., some previous work is worth citing and discussing, including (but not limited to):

- Automated monitoring of animal behaviour with barcodes and convolutional neural networks. Tim Gernat, Tobias Jagla, Beryl M Jones, Martin Middendorf, and Gene E Robinson. bioRxiv posted 27 November 2020 doi:10.1101/2020.11.27.401760.
- Boenisch, Franziska et al. (2018). "Tracking All Members of a Honey Bee Colony Over Their Lifetime Using Learned Models of Correspondence". In: *Frontiers in Robotics and AI* 5, p. 35.

2) I see the paper's main contribution as providing the first elements of a tool that might in the future be used by other groups working on collective behavior. Therefore, the authors should provide enough information to allow potential users to assess whether BACH is appropriate for their research system and questions, and if so, to easily test it. With this in mind, some important information is currently missing, hard to locate, or presented in insufficient details:

- As far as I can tell, there is little information provided on Github on how to start using the software (the ReadMe file appears to be minimal). What are the minimum requirements, supported operating systems, etc.? Will BACH run on a computing cluster? All that information should be available to potential users.
- It is crucial for potential future users to understand the relative performance of BACH compared to existing systems. In that respect, a table summarizing these differences in quantitative terms (e.g., rate of missed ants, and false positives, etc. for BACH and other tracing systems) would be useful to get a rapid, "compact" overview.

- It would be useful for the reader to get a sense of the size of the tags relative to the ants, of the size of the ants relative to the setup, and of the walking speed of the ants (as a basis of comparison with their study species); I was surprised by the lack of example video of ants moving about the setup (I understand the tracking algorithm doesn't save videos, but presumably, recording such a video separately would be reasonably straightforward). Instead of showing the setup (Fig. 1a-c) and ants (Fig. 1e) separately, could the authors show a short supplementary video of an actual experiment, or at least an image of the setup with tagged ants in it? If Fig. 1e stays in the MS, it should also have a scale bar.
- Recording: what image size was used?

Minor comments:

L. 97: Why was the tunnel kept closed? Presumably, its function is to allow movement between the nest and foraging arena and keeping it closed would prevent normal colony behavior.

L. 121: I was surprised to see the authors tagged the abdomen instead of the thorax of the ants, because the abdomen can bend underneath the body and the tag become invisible when this happens. Can the authors explain why they chose this unusual method?

L. 128: Why were non-tagged ants included in the training set? That seems like something that might decrease the performance of the algorithm.

L. 172: This section would benefit from being placed earlier in Material and methods (before the description of BACH), along with all the technical information regarding recording (frame rate, image size)

L. 436: The main advantages are not entirely convincing to me. The first advantage is also implemented

I wonder to what extent the performance of BACH was affected by the behavior of the ants. It looks like ants were stressed (recently moved to a new, sometimes brightly lit arena), and were not given much time to settle. Can the authors discuss how BACH would be expected to perform under more natural conditions (larger colonies, presence of a queen, natural foraging instead of placing ants in the foraging arena and preventing them from entering the nest)? My feeling is that some aspects might be improved (e.g. because the average walking speed might be lower) but other could be made much more challenging (e.g., clustering of ants and occlusion of tags will increase in the nest with "real" colonies).

L. 328: the surprisingly low performance of human observer makes me wonder if the performance of the tracking algorithm is not primarily constrained by poor video quality (i.e. hardware limitation) in the first place.

L. 382: I'm not sure I understand the claim that BACH's real-time performance will not be compromised. Can the authors develop the arguments? What is meant by "module"?

L. 436: I'm not sure I understand and agree with the claim that BACH's main advantage compared to other tracking systems is that it "remembers the identities of individuals even when their identity codes become invisible, and can successfully do so as long as it continuously detects their shape". As far as I understand this sentence, antTrax does exactly this: it assigns IDs to shape-based trajectories, even when the individual tag is not visible in all frames in the trajectory. As noted above (see main comment), I'm also not sure about the second claim ("BACH generates data in real time through relatively light computational processes"). The light computational

processes might indeed be a good argument, but in that case, I would back this claim with numbers and quantitative comparisons with other software. Generally, I would encourage the authors to put less emphasis on claims of novelty.

Reviewer: 2

Comments to the Author(s)

General comments:

The authors describe a newly developed real-time ant tracking system that uses a CNN to detect ants, and a combination of barcodes and a unique and lightweight tracking algorithm to identify them. They evaluate their tracking system in a pleasingly thorough and rigorous way, and highlight avenues for improvement.

My main concern relates to the novelty of their approach. Real-time social insect tracking systems already exist. For example, the very first ant tracking system by Mersch et al. (2013) processes its videos in real time, using only barcodes. More recently, Wild et al. (2018) described a system that uses barcodes and a CNN to track honey bees in real time. I suggest the authors compare their tracking system to existing real time tracking systems and highlight how this work advances the field. This does not require new data; it could be done verbally in the Discussion.

Secondly, current social insect tracking systems are able to reliably follow >1,000 individuals. It is fine to evaluate a tracking system with just 20 individuals, as was done here, but then the manuscript should either state that the system is designed for small colonies or it should extrapolate how well critical components (i.e., the CNN for detecting the ants and the algorithm for maintaining identities when the barcode cannot be read) will perform when more individuals are being tracked.

Specific comments:

Lines 1-2: The title is slightly misleading, because BACH tracks ant identities and positions, not behavior. I suggest modifying this to "Integrating real-time data analysis into automatic social insect tracking".

Lines 19-20: Reading the abstract the first few times I found the phrase "concerning computer vision-based ant detection only" confusing. It only made sense to me after reading the main text. I suggest to delete this.

Lines 38-39: The discussion would benefit from a slightly broader perspective. I suggest to change "ants" to "social insects" and cite some of the honey bee tracking systems (e.g. Landgraf et al. 2015 *Frontiers in Robotics and AI* and Gernat et al. 2018 in *PNAS*).

Lines 39-41: The behavioral repertoire of social insects is commonly thought to be rich, not narrow, and their social interactions are complex, not simple. I suggest to change this sentence accordingly.

Lines 41-42: If it was easy to scrape a significant amount of information just by scanning images, automatic trackers would do a better job at this. I suggest to delete "easy".

Lines 50-53: For completeness sake, and as a pointer to the next "next generation", this should also cite Gernat et al. 2020 on bioRxiv, which uses AI to detect honey bee behavior.

Lines 54-68: The first sentence of this paragraph should cite the tracking systems that rely on offline processing. More importantly, though, the authors should mention which systems are already capable to process their footage in real-time, such as Wild et al. 2018 on arXiv and Mersch et al. 2013 in Science.

Lines 82-83: The distinction between efficiency and performance is unclear. Moreover, lines 78-79 suggest that BACH is much more efficient than HO, which seemingly contradicts this statement.

Lines 175-177: Please specify the video resolution, so BACH can be better compared to other real-time capable tracking systems, and so it is possible to approximate the size of the barcode in the videos.

Line 181: The abbreviation "HO" is defined at multiple places in the manuscript. Defining it once would be sufficient.

Line 322: I was unable to find Table S1.

Lines 393-395: A better approach would be to use brighter lights so the camera will use a lower exposure time, which will result in less motion blur. Using a higher frame rate will not help to increase the detection rate.

Lines 479-480: It is unclear what the error bars represent.

References:

Mersch, D. P., Crespi, A. & Keller, L. Tracking individuals shows spatial fidelity is a key regulator of ant social organization. *Science* 340, 1090-3 (2013).

Wild, B., Sixt, L. & Landgraf, T. Automatic localization and decoding of honeybee markers using deep convolutional neural networks. 1-20 (2018).

===PREPARING YOUR MANUSCRIPT===

===PREPARING YOUR REVISION IN SCHOLARONE===

Author's Response to Decision Letter for (RSOS-202033.R0)

See Appendix A.

Decision letter (RSOS-202033.R1)

Dear Dr Teseo

On behalf of the Editors, we are pleased to inform you that your Manuscript RSOS-202033.R1 "Integrating real-time data analysis into automatic tracking of social insects" has been accepted for publication in Royal Society Open Science subject to minor revision in accordance with the referees' reports. Please find the referees' comments along with any feedback from the Editors below my signature.

Please submit your revised manuscript and required files (see below) no later than 7 days from today's (ie 23-Feb-2021) date. Note: the ScholarOne system will 'lock' if submission of the revision is attempted 7 or more days after the deadline. If you do not think you will be able to meet this deadline please contact the editorial office immediately.

on behalf of Dr Agustina Gómez-Laich (Associate Editor) and Kevin Padian (Subject Editor)
 openscience@royalsociety.org

Associate Editor Comments to Author (Dr Agustina Gómez-Laich):
 Associate Editor

Comments to the Author:

I have some few minor comments and suggestions that are listed below. Specific comments relate to the page and line number of the clean version of the word document that was available for review.

Introduction.

Line 72-81. This paragraph presents results, thus I suggest eliminating it from the introduction.

Methods and results.

Check decimals. The number of decimals presented is not consistent along the Ms.

Check \pm simbol along the Ms.

Discussion.

Line 414-415. Can you please rephrase this sentence. It is not clear.

Line 430-432. Can you please rephrase this sentence. It is not clear.

Line 492-500. Please incorporate references that support the statements.

Figure captions.

There is no reference to figure 4 in the text.

===PREPARING YOUR MANUSCRIPT===

While not essential, it will speed up the preparation of your manuscript proof if you format your references/bibliography in Vancouver style (please see

<https://royalsociety.org/journals/authors/author-guidelines/#formatting>). You should include DOIs for as many of the references as possible.

===PREPARING YOUR REVISION IN SCHOLARONE===

<https://royalsociety.org/journals/authors/author-guidelines/#data>. You should ensure that you cite the dataset in your reference list. If you have deposited data etc in the Dryad repository,

please only include the 'For publication' link at this stage. You should remove the 'For review' link.

Author's Response to Decision Letter for (RSOS-202033.R1)

See Appendix B.

Decision letter (RSOS-202033.R2)

Dear Dr Teseo,

It is a pleasure to accept your manuscript entitled "Integrating real-time data analysis into automatic tracking of social insects" in its current form for publication in Royal Society Open Science.

Please see the Royal Society Publishing guidance on how you may share your accepted author manuscript at <https://royalsociety.org/journals/ethics-policies/media-embargo/>. After

publication, some additional ways to effectively promote your article can also be found here <https://royalsociety.org/blog/2020/07/promoting-your-latest-paper-and-tracking-your-results/>.

on behalf of Dr Agustina Gómez-Laich (Associate Editor) and Kevin Padian (Subject Editor)
openscience@royalsociety.org

Appendix A

Royal Society Open Science - Decision on Manuscript ID RSOS-202033

Flag for follow up.

Royal Society Open Science <onbehalf@manuscriptcentral.com>

Thu 1/28/2021 1:32 AM

Dear Dr Teseo

The Editors assigned to your paper RSOS-202033 "Integrating real-time data analysis into automatic tracking of social insect behaviour" have now received comments from reviewers and would like you to revise the paper in accordance with the reviewer comments and any comments from the Editors. Please note this decision does not guarantee eventual acceptance.

Please submit your revised manuscript and required files (see below) no later than 21 days from today's (ie 27-Jan-2021) date. Note: the ScholarOne system will 'lock' if submission of the revision is attempted 21 or more days after the deadline. If you do not think you will be able to meet this deadline please contact the editorial office immediately.

Best regards,

Lianne Parkhouse

Editorial Coordinator

on behalf of Dr Agustina Gómez-Laich (Associate Editor) and Kevin Padian (Subject Editor)
openscience@royalsociety.org

Associate Editor Comments to Author (Dr Agustina Gómez-Laich):

The manuscript entitled “Integrating real-time data analysis into automatic tracking of social insect behaviour” has now been seen by two reviewers. Both of them found the work interesting and commend the authors for developing a new tool for automatic video tracking. However, reviewers raised several concerns and requested some substantial changes. One of the principal concerns both reviewers have is related to the novelty of the approach. Authors emphasize that most current systems rely on offline data analysis instead of tracking insects on a real time basis and present BACH as a novelty to address this gap however, one of the first tracking systems used in ants several years ago (2013) already implemented online image analyses. Thus, both reviewers recommend authors to discuss more in detail about other systems, compare BACH to existing real time tracking systems, highlight the benefits of using this new tool and put less emphasis on claims of novelty. In addition, reviewer#1 considers that important information is currently missing, hard to locate, or presented in insufficient details. This reviewer suggest authors to give more detailed information about how BACH works (e.g. minimum requirements, supported operating systems) and include a short supplementary video or image of the setup with tagged ants in it to allow the reader to get a sense of the size of the tags, of the setup and of the walking speed of the ants. With this information potential users would be able to evaluate if this tool is appropriate for their research system and test it. Finally, another major concern both reviewers have is how BACH would be expected to perform on larger colonies and under more natural conditions.

We thank the Editor for the suggestions. Accordingly, we have now addressed all the Reviewers' comments, largely amplifying our discussion with comparisons to other similar tracking systems. We have toned down our claims of novelty across the manuscript and included references to detailed code instructions, models and videos that we uploaded online (Github and Zenodo). We have emphasized the fact that we only tested small groups of ants and not real colonies in natural conditions. Using the Reviewers' advice, we made predictions about BACH's potential performance in experiments addressing biologically relevant hypotheses. Please note that line numbers refer to the manuscript including highlighted changes.

Reviewer comments to Author:

Reviewer: 1
Comments to the Author(s)

This paper presents a new, custom-made system for automated behavioral tracking in ants. Automated behavioral tracking and machine learning for behavioral analyses is an exploding field

of research, and BACH is a welcome addition to the growing list of such tools. However, I have several comments, concerns, and questions regarding the manuscript, listed below, most of which can be addressed without performing additional experiments.

We thank Reviewer 1 for their constructive comments, all of which we have carefully addressed. Please note that line numbers refer to the manuscript including highlighted changes.

Major comments:

1) The paper's main claim of novelty is based on its use of real-time image analysis, but one of the first tracking systems used in ants already implemented online image analyses (Mersch, Crespi, Keller, 2013. Science). The claim that "most current systems rely on offline data analysis" (and the focus on this aspect throughout the manuscript) therefore seems overstated.

Also in response to the comments of Reviewer 2, we now consider Mersch et al 2013 as a real-time tracker, referenced at line 67 among systems using real time data analysis. We had previously considered Mersch et al 2013 as a purely offline tracker because the online feature is only mentioned once in the supplementary materials of the manuscript. We thank both Reviewer 1 and 2 for letting us notice.

We have now toned down the abstract and introduction claims that 'most' systems rely on offline data analysis, and state that this is rather an ongoing process. More specifically:

- In the abstract, we have replaced 'most' with 'many' (Line 11), and added 'further' before 'integrating real time data analysis...' (line 23).
- In the introduction, we have modified the previous statement:

'Although the technology underlying insect video tracking advances rapidly and significantly, most of the current systems still rely on offline data processing (references). This means that researchers often perform tracking on previously recorded videos and analyse data only *a posteriori*.'

into:

'Although the technology underlying insect video tracking advances rapidly and significantly, many recently developed systems rely on offline data processing (references). This means that researchers often perform tracking on previously recorded videos and analyse data *a posteriori*.' (Lines 52-54).

- In the introduction, we have modified the previous statement:

'Graphics Processing Units (GPUs), which allow accelerating computation, have instead the potential to address this gap via integrating real time data analysis^{23,24} in the traditional way of conducting tracking-based studies on insect behavior^{12,25}.'

into:

‘Graphics Processing Units (GPUs), which allow accelerating computation, begin to address this gap via integrating real time data analysis^{23,24} in the traditional way of conducting tracking-based studies on insect behavior^{12,25}. ‘ (Lines 65-67)

Please note that I don’t feel novelty is essential, as several tracking systems have been developed in parallel in different labs over the last decade, and the results of those efforts are worth sharing, irrespective of any redundancy in function.

Along the same lines, the authors focus heavily on the comparison between BACH and Antrax, while largely ignoring earlier, more similar (barcode based tracking in infra-red light) systems. Can the authors discuss other tracking systems more broadly? Beyond Mersch et al., some previous work is worth citing and discussing, including (but not limited to):

- Automated monitoring of animal behaviour with barcodes and convolutional neural networks. Tim Gernat, Tobias Jagla, Beryl M Jones, Martin Middendorf, and Gene E Robinson. bioRxiv posted 27 November 2020 doi:10.1101/2020.11.27.401760.
- Boenisch, Franziska et al. (2018). “Tracking All Members of a Honey Bee Colony Over Their Lifetime Using Learned Models of Correspondence”. In: *Frontiers in Robotics and AI* 5, p. 35.

We have now expanded our discussion to include references to these and other tracking systems, also addressing similar comments from Reviewer 2. The following paragraph is about Boenisch et al *Front Robotics* 2018 and Wild et al *ArXiv* 2018 (these studies are highly interconnected) at lines 534-540, as follows:

‘Another recent tracker for honeybees identifies individual custom-made tags encoding the identity of bees, also retrieving information about the bee body orientation via the tag inclination^{25,37}. It combines consecutive individual detections into short reliable tracklets and then connects them over longer gaps, using machine learning at both steps. This allows tracking back individual trajectories that also include frames with undetectable tags. While the tag recognition portion of this tracker can in principle run in real time through the use of GPUs, the authors still used multiple nodes of a large cluster to track honeybees offline in their experiments.

Then, we had already mentioned Gernat 2018 and another 2020 manuscript using this system (Geffre et al 2020). Gernat 2020 was uploaded on BiorXiv after we submitted the present manuscript. However, we agree with Reviewer 1 that this manuscript is highly relevant to our work, and expanded the discussion related to it, also referring to the recent BiorXiv preprint. In particular, we have now modified the previous paragraph:

‘Other automatic trackers used to study insect societies integrate computer vision and/or matrix code identification to different extents, working mainly offline. For example, a recent system for tracking honeybee colonies not only uses matrix codes for individual identification, but also allows automatically and simultaneously detecting multiple mouth to mouth food exchanges

(trophallaxis). The system identifies these episodes based on the relative position and orientation of matrix codes, confirming such identifications via custom computer vision algorithms.'

Into an expanded version (Lines 514-533):

'Other automatic trackers used to study insect societies integrate computer vision and/or matrix code identification to different extents, working mainly offline. For example, a recent system for tracking honeybee colonies^{17,19,20} not only uses matrix codes for individual identification, but also allows automatically and simultaneously detecting multiple mouth to mouth food exchanges (trophallaxis), simultaneously tracking high numbers of individuals (>1000). The system, which similar to BACH integrates computer vision and matrix code identification, classifies these episodes based on the relative position and orientation of matrix codes, confirming such identifications via custom computer vision algorithms. However, it does not remember the identities of individuals for which matrix codes become invisible. This requires confining experimental honeybees in a relatively narrow space that prevents them from turning or forming multi-layer clusters where individuals on upper layers cover the markers of individuals in lower layers.'

2) I see the paper's main contribution as providing the first elements of a tool that might in the future be used by other groups working on collective behavior. Therefore, the authors should provide enough information to allow potential users to assess whether BACH is appropriate for their research system and questions, and if so, to easily test it. With this in mind, some important information is currently missing, hard to locate, or presented in insufficient details:

- As far as I can tell, there is little information provided on Github on how to start using the software (the README file appears to be minimal). What are the minimum requirements, supported operating systems, etc.? Will BACH run on a computing cluster? All that information should be available to potential users.

In response to Reviewer 1's comments, we have now added the command lines we used in the study, with instructions, to the file README in the BACH Github. The file also includes the requirements for using BACH. We have uploaded on Zenodo the two tracking models we used, to which we refer at lines 175-177.

- It is crucial for potential future users to understand the relative performance of BACH compared to existing systems. In that respect, a table summarizing these differences in quantitative terms (e.g., rate of missed ants, and false positives, etc. for BACH and other tracing systems) would be useful to get a rapid, "compact" overview.

We agree with Reviewer 1 that a quantitative comparison of the relative performance of BACH compared to the other systems would be ideal. Overall, at least three recent studies on trackers provide details about the accuracy of their systems (some even compare previous versions and/or different data sets, like the Kronauer group study on eLife). Similarly, self evaluation is one of the priorities of our study on BACH.

Nevertheless, providing a truly informative compact table illustrating inter-system efficiency differences is not trivial. This has been to some extent attempted in figures 4.1 and 4.2 of Wild et al 2018 (ArXiv). However, the authors could provide a thorough comparison only with a previous study from the same laboratory (Wario et al 2015). We believe that a series of confounding interacting factors complicate a hypothetical inter-system comparison. First, each system differs from the others from a functional perspective. Tag detectors vary among studies, systems may or may not use tracklets to disentangle trajectories, may work online, offline or both, with or without implementing error correction algorithms. Second, systems largely differ due to the diversity of their hardware settings and tracking conditions, which contributes significantly to the efficiency measures proposed in the respective studies. For example, while several groups using matrix or similar tags employ high-definition industrial cameras (Wild et al 2018, Boenisch et al 2018, Gernat et al 2018, 2020), we used here standard webcams modified to become IR-light sensitive. As a result, while the ArUco code detector itself has in principle very high efficiency (Garrido-Jurado et al 2014,2016), it fails to reach its standards in our study most probably because the quality of our images is relatively low.

Nonetheless, we found the Reviewer 1's comment appropriate and constructive, and we have now added a discussion paragraph that first elaborates on the above-mentioned ideas on quantitative comparisons, and then connects with a paragraph about qualitative differences between BACH and other real time trackers. As follows (Lines 547-565):

'While our work on BACH and other studies have provided self evaluations of the respective systems' efficiency, several interacting factors complicate quantitative comparisons among different trackers. First, systems largely differ due to the diversity of their hardware settings and tracking conditions, which contributes to their efficiency. For example, several studies rely on high-definition industrial cameras (Wild et al 2018, Boenisch et al 2018, Gernat et al 2018, 2020), whereas here we only employed standard webcams modified to become IR-light sensitive. These produced relatively low-quality images and low FPS videos. As a result, while the original standalone ArUco code detector has in principle a very high efficiency²⁸, it failed to reach such standards in our study. Secondly, each tracker differs from the others from a software perspective, both in function and scope. For instance, different research groups employ different matrix code identifiers, sometimes developing their own algorithms. Some studies integrate the identification of codes with computer vision, while others apply convolutional neural networks to matrix code identification. Different systems work online, offline or both, in some cases relying on a *posteriori* tracklet integration for disentangling individual trajectories. Finally, while some groups have conceived their systems to achieve maximal accuracy regardless of the computational effort involved, the current version of BACH targets rapid data harvesting and sharing to the expense of trajectory reconstruction accuracy and cluster solving. A systematic review evaluating and comparing systems' capabilities would therefore require a significant amount of non-trivial considerations.'

- It would be useful for the reader to get a sense of the size of the tags relative to the ants, of the size of the ants relative to the setup, and of the walking speed of the ants (as a basis of comparison with their study species);

We have now integrated sentences at lines 138-140 with mean plus sd measures of the ant lengths, as well as relative size of ArUco codes sizes compared to ants. The walking speed of ants in different conditions is given in the result section at lines 321-331.

I was surprised by the lack of example video of ants moving about the setup (I understand the tracking algorithm doesn't save videos, but presumably, recording such a video separately would be reasonably straightforward). Instead of showing the setup (Fig. 1a-c) and ants (Fig. 1e) separately, could the authors show a short supplementary video of an actual experiment, or at least an image of the setup with tagged ants in it?

We have eliminated panel e from figure 1. We have now uploaded on Zenodo three of the videos we used for experiments, one for each of our three tracking conditions, providing a DOI for the videos at line 173.

If Fig. 1e stays in the MS, it should also have a scale bar.

We have now removed Fig. 1e and accordingly modified references to Fig. 1 throughout the manuscript.

- Recording: what image size was used?

To respond to both Reviewer 1 and Reviewer 2 comments, we have now modified the text at line 168, integrating a sentence about the image size:

'We recorded videos at 10 frames/second for 25 seconds and with a resolution of 4096 x 2160 pixels, in three different conditions.'

Minor comments:

L. 97: Why was the tunnel kept closed? Presumably, its function is to allow movement between the nest and foraging arena and keeping it closed would prevent normal colony behavior.

In this study, we aimed to conduct preliminary experiments to assess the efficiency of BACH. To keep conditions as simple as possible, we kept the tunnel closed and focused on a single area per experimental session. To clarify this point, we have integrated the following sentence at lines 108-111:

'As this study aimed to assess BACH's efficiency, and not to test biological hypotheses, we decided to focus on a single area per experimental session. Therefore, we kept the tunnel closed to prevent ants from walking between the foraging and nest areas.'

L. 121: I was surprised to see the authors tagged the abdomen instead of the thorax of the ants, because the abdomen can bend underneath the body and the tag become invisible when this happens. Can the authors explain why they chose this unusual method?

Even though some studies tag ants on the abdomen rather than the thorax (e.g. Quque et al 2020 Insect Science, DOI 10.1111/1744-7917.12792), we agree with Reviewer 1 that our approach leads to identification problems when ants bend the abdomen underneath the body. However, applying glue on the thorax, which we did in some preliminary tests, occasionally resulted in ants using legs to remove their codes, and/or in legs becoming irreparably entangled in the glue. Since this study aimed to test our tracking device rather than studying biological hypotheses, we decided to opt for the safest, fastest approach. The abdomen offers a larger surface that simplifies the glueing procedure and minimizes risks of damage for the ants. We have now integrated the following sentence at lines 140-143 in order to explicit our reasoning:

‘Researchers often mark ants on the thorax because sometimes these bend their abdomen underneath the body, hiding tags from the camera view. However, after conducting preliminary tests on thorax-tagged ants, we opted for abdomen tagging as it prevented ants from removing codes with their legs, and legs from becoming irreparably entangled in the glue.’

L. 128: Why were non-tagged ants included in the training set? That seems like something that might decrease the performance of the algorithm.

We agree with Reviewer 1 that the algorithm may not benefit from including non-tagged ants. However, we decided to include non-marked ants mainly for two reasons: 1) to be able to track ants even in case they lost their codes, or in case their codes became invisible to the camera for some reason; 2) to prevent the algorithm from relying on the code itself, for example via recognizing specific codes and not others. To explicit our reasoning, we have now included the following sentence at lines 150-153:

‘We decided to include images of non-tagged ants in order to track them even in case they lost their codes, or in case their codes became invisible to the camera for some reason. In addition, this prevented the algorithm from relying on codes themselves, and for example learning to identify some specific codes and not others.’

L. 172; This section would benefit from being placed earlier in Material and methods (before the description of BACH), along with all the technical information regarding recording (frame rate, image size)

We have now moved this section above, immediately after the description of the model training and before the description of BACH (Lines 164-173).

L. 436: The main advantages are not entirely convincing to me. The first advantage is also implemented

As the sentence is not complete and points are raised by Reviewer 1 below regarding the same line 436, we interpret this as a typo and reply to the comments below.

I wonder to what extent the performance of BACH was affected by the behavior of the ants. It looks like ants were stressed (recently moved to a new, sometimes brightly lit arena), and were not given much time to settle. Can the authors discuss how BACH would be expected to perform under more natural conditions (larger colonies, presence of a queen, natural foraging instead of placing ants in the foraging arena and preventing them from entering the nest)? My feeling is that some aspects might be improved (e.g. because the average walking speed might be lower) but other could be made much more challenging (e.g., clustering of ants and occlusion of tags will increase in the nest with “real” colonies).

We agree with the points raised by Reviewer 1, and we have therefore added a new paragraph at lines 476-488. As follows:

‘Regarding the general validity of our results, we must stress that our experimental conditions differ significantly from those required in biologically relevant experiments. For example, *Camponotus* colonies usually include hundreds or thousands of workers (and one or multiple queens), whereas we only tested 20-individual queenless experimental groups. In addition, we tested individuals in visible light conditions that they most likely encounter only rarely in natural conditions, which resulted in stress and high walking speeds. Conducting long-term experiments in biologically sound settings would probably trigger a significantly different set of behaviors. For example, we would expect ants to avoid foraging in visible light, and in general to walk relatively slower due to lower stress levels; we would also expect higher levels of clustering due to higher densities of individuals within the nest. Therefore, ants walking at a lower average speed would in principle improve BACH’s identification efficiency, but their increased tendency to cluster would probably increase multiple detections. Further tests with larger colonies and a realistic day/night alternation in the foraging area will tell how BACH performs its tracking in more natural conditions.’

L. 328: the surprisingly low performance of human observer makes me wonder if the performance of the tracking algorithm is not primarily constrained by poor video quality (i.e. hardware limitation) in the first place.

We agree with Reviewer 1 that the performance of the human observer is low and most probably depends also on hardware limitations. We mentioned this already in the previous version of the manuscript, lines 393-399. In our study, instances of non-identified ants occurred when the codes were very blurry because of high walking speeds, mostly in the foraging area. In order to clarify (and to respond to a point Reviewer 2 raised), we have now integrated the following statement and modified a sentence immediately afterwards. As follows:

‘The fact that even HO showed a relatively low identification accuracy points first of all to the need for upgrading our hardware settings.’ (Lines 455-456)

'In future work, we aim to increase light intensity to shorten cameras' exposure times, reducing the occurrence of blurry images and images and therefore maximizing code readability.' (456-466)

Finally, we address the relatively low quality of our images in a new paragraph integrated to respond to another Reviewer 1's comment, as follows (Lines 550-554):

'For example, several studies rely on high-definition industrial cameras (references), whereas here we only employed standard webcams modified to become IR-light sensitive. This produced relatively low-quality images and low FPS videos. As a result, while the original standalone ArUco code detector²⁸ has in principle a very high efficiency, it failed to reach such standards in our study.'

L. 382: I'm not sure I understand the claim that BACH's real-time performance will not be compromised. Can the authors develop the arguments? What is meant by "module"?

We agree that this statement needed further clarification, and therefore we have now included a few sentences that develop the module definition, as Reviewer 1 suggests (Lines 439-444):

'BACH runs different units of code implementing its different features in parallel and independently, running them via either separate processes or threads. In this way, modules with higher computational complexity do not block modules with lower computational complexity but stricter real-time requirements, such as the tracker. Therefore, implementing further modules to take into account trajectory histories will in principle not compromise its real-time performance.'

L. 436: I'm not sure I understand and agree with the claim that BACH's main advantage compared to other tracking systems is that it "remembers the identities of individuals even when their identity codes become invisible, and can successfully do so as long as it continuously detects their shape". As far as I understand this sentence, antTrax does exactly this: it assigns IDs to shape-based trajectories, even when the individual tag is not visible in all frames in the trajectory. As noted above (see main comment), I'm also not sure about the second claim ("BACH generates data in real time through relatively light computational processes"). The light computational processes might indeed be a good argument, but in that case, I would back this claim with numbers and quantitative comparisons with other software. Generally, I would encourage the authors to put less emphasis on claims of novelty.

Following the point raised by Reviewer 1, we have now significantly toned down the paragraph by referring only to other real time trackers, and by further eliminating other claims of novelty. We have modified the paragraph from the previous version of the manuscript:

'Compared to other tracking systems, BACH has two main advantages. First, it remembers the identities of individuals even when their identity codes become invisible, and can successfully do so as long as it continuously detects their shape. Secondly, and most importantly, BACH

generates data in real time through relatively light computational processes, which constitutes its main novelty and strength. Although BACH needs to undergo upgrading from several perspectives, we developed it with the goal of emphasizing the possibility for real time data to ameliorate animal behaviour research that relies on video tracking. Real time data shortens the delay between observation and data generation to virtually non-existent, enabling researchers to alter experimental conditions based on current results.'

to :

'From a qualitative perspective, BACH has two main differences compared to other real time tracking systems. First, it remembers the identities of individuals even when their identity codes become invisible, and can successfully do so as long as it continuously detects their shape. Secondly, and most importantly, it generates real time data through relatively light computational processes. Although BACH needs to undergo upgrading from several perspectives, we developed it with the goal of implementing real time data analysis and ameliorating animal behavior research that relies on video tracking. Real time data analysis shortens the delay between observation and result generation to virtually non-existent, enabling researchers to alter experimental conditions based on current results.' (Lines 565-573)

We also agree with Reviewer 1 that the pointed sentence was to some extent misleading. In fact, the outcome of what BACH does is similar to the outcome of what AnTrax does, but the way they do it is different. When individuals x and y form a cluster, AnTrax knows that x and y are there but does not know exactly where they are. After x and y break free from the cluster, AnTrax retrieves their identity and keeps tracking them. Contrarily, BACH tries to identify individuals at all times, regardless of whether they are alone or in clusters. This function relies on the principle that, if the camera works at high FPS levels, an individual cannot move too far from its position in the previous frame. Therefore, BACH tries to associate the identity of previous detections to current detections. In principle, BACH could tell the identity of individuals even within clusters. However, it should rely on high quality images that we have not produced in the current study.

Given that the paragraph previously starting at line 436 has now changed and refers to the differences between BACH and other real time trackers (and not to all trackers), we have explicit the concepts mentioned here at lines 432-444, when AnTrax is first mentioned:

'Although BACH doesn't have a specific cluster-solving function, it tries to identify individuals at all times, regardless of whether they form clusters or not. This function relies on the principle that, if the camera works at high FPS settings, an individual cannot move too far from its position in the previous frame. BACH therefore tries to associate the identity of previous detections to current detections and, at least in principle, could tell the identity of individuals even within clusters. However, doing this with high efficiency would require high quality images that we have not produced in the current study. In future versions of BACH, we aim to increase image quality and integrate a similar process using individual histories to retain ant identities while they engage in clusters. In principle, as BACH runs different units of code implementing its different features in parallel and independently, running them via either separate processes or threads. In this way,

modules with higher computational complexity do not block modules with lower computational complexity but stricter real-time requirements, such as the tracker. Therefore, implementing further modules to take into account trajectory histories will in principle not compromise its real-time performance.'

To adapt the text to the latter paragraph, we have deleted 'Although we still need to implement a solution for this in BACH', lines 421-422 of the previous version, and 'This system, optimized for cluster-forming Clonal Rider Ants' (previous manuscript line 376). We now state that AnTrax focuses on Clonal Raider Ants at line 510:

'This does not affect virtually blind Clonal Raider Ants, on which AnTrax specializes, but...'

Reviewer: 2

Comments to the Author(s)

General comments:

The authors describe a newly developed real-time ant tracking system that uses a CNN to detect ants, and a combination of barcodes and a unique and lightweight tracking algorithm to identify them. They evaluate their tracking system in a pleasingly thorough and rigorous way, and highlight avenues for improvement.

My main concern relates to the novelty of their approach. Real-time social insect tracking systems already exist. For example, the very first ant tracking system by Mersch et al. (2013) processes it's videos in real time, using only barcodes. More recently, Wild et al. (2018) described a system that uses barcodes and a CNN to track honey bees in real time. I suggest the authors compare their tracking system to existing real time tracking systems and highlight how this work advances the field. This does not require new data; it could be done verbally in the Discussion.

We thank Reviewer 2 for their constructive criticism that we believe allowed us to improve the quality of our manuscript. We have now toned down our previous claims of novelty, also in response to Reviewer 1's comments. Our discussion now includes comparisons with tracking systems pointed by Reviewers 1 and 2.

Please note that line numbers refer to the manuscript including highlighted changes.

Secondly, current social insect tracking systems are able to reliably follow >1,000 individuals. It is fine to evaluate a tracking system with just 20 individuals, as was done here, but then the manuscript should either state that the system is designed for small colonies or it should extrapolate how well critical components (i.e., the CNN for detecting the ants and the algorithm for maintaining identities when the barcode cannot be read) will perform when more individuals are being tracked.

According to Reviewer 2's point, we have now integrated a discussion paragraph (also in response to a Reviewer 1's comment, lines 476-488) where we state that in this study we only tracked 20-individual groups, and tracking real colonies would likely result in a whole new set of behavior that BACH will have to deal with. As follows:

'Regarding the general validity of our results, we must stress that our experimental conditions differ significantly from those required in biologically relevant experiments. For example, *Camponotus* colonies usually include hundreds or thousands of workers (and one or multiple queens), whereas we only tested 20-individual queenless experimental groups. In addition, we tested individuals in visible light conditions that they most likely encounter only rarely in natural conditions, which resulted in stress and high walking speeds. Conducting long-term experiments in biologically sound settings would probably trigger a significantly different set of behaviors. For example, we would expect ants to avoid foraging in visible light, and in general to walk relatively slower due to lower stress levels; we would also expect higher levels of clustering due to higher densities of individuals within the nest. Therefore, ants walking at a lower average speed would in principle improve BACH's identification efficiency, but their increased tendency to cluster would probably increase multiple detections. Further tests with larger colonies and a realistic day/night alternation in the foraging area will tell how BACH performs its tracking in more natural conditions.'

Specific comments:

Lines 1-2: The title is slightly misleading, because BACH tracks ant identities and positions, not behavior. I suggest modifying this to "Integrating real-time data analysis into automatic social insect tracking".

Following the suggestion of Reviewer 2, we have now modified the title to "Integrating real-time data analysis into automatic tracking of social insects".

Lines 19-20: Reading the abstract the first few times I found the phrase "concerning computer vision-based ant detection only" confusing. It only made sense to me after reading the main text. I suggest to delete this.

We agree with Reviewer 2 that that sentence was misleading in the abstract. However, we still wanted to specify that there are two different processes going on, ant detection and ant identification. Therefore, we have now modified the sentence to:

'We found that BACH detected ant shapes only slightly worse than the human observer. However, its matrix code-mediated identification of individual ants only attained human-comparable levels when ants moved relatively slowly, and fell when ants walked relatively fast. ' (Lines 17-19)

Lines 38-39: The discussion would benefit from a slightly broader perspective. I suggest to change "ants" to "social insects" and cite some of the honey bee tracking systems (e.g. Landgraf et al. 2015 *Frontiers in Robotics and AI* and Gernat et al. 2018 in *PNAS*).

We believe Reviewer 2 refers here to Boenisch et al 2018 (Landgraf as last author) *Frontiers in Robotics and AI*. We have now integrated paragraphs referring to these and other studies in the discussion.

We have now replaced 'ants' with 'insects' in a series of relevant instances within the discussion (lines 506 and 508).

Lines 39-41: The behavioral repertoire of social insects is commonly thought to be rich, not narrow, and their social interactions are complex, not simple. I suggest to change this sentence accordingly.

This sentence was conceived with in mind the behavioral complexity of other animals used in laboratory behavioral experiments, such as mice. However, we agree with Reviewer 2 that the sentence reads odd without an explicit comparison with vertebrates. On the other hand, this would be out of place in this context. We have therefore decided to remove the sentence.

Lines 41-42: If it was easy to scrape a significant amount of information just by scanning images, automatic trackers would do a better job at this. I suggest to delete "easy".

We have now deleted the word easily as suggested by Reviewer 2. (Line 40)

Lines 50-53: For completeness sake, and as a pointer to the next "next generation", this should also cite Gernat et al. 2020 on bioRxiv, which uses AI to detect honey bee behavior.

The reference to Gernat 2020 has been integrated here (line 53) and where needed across the manuscript. We had not mentioned it in our manuscript as it appeared on BiorXiv after we submitted our manuscript to R Soc Open Science.

Lines 54-68: The first sentence of this paragraph should cite the tracking systems that rely on offline processing. More importantly, though, the authors should mention which systems are already capable to process their footage in real-time, such as Wild et al. 2018 on arXiv and Mersch et al. 2013 in *Science*.

We have now added references, as Reviewer 2 suggests, in relation to both offline and online trackers in the first and second sentence of the paragraph, respectively (Lines 53 and 67).

Lines 82-83: The distinction between efficiency and performance is unclear. Moreover, lines 78-79 suggest that BACH is much more efficient than HO, which seemingly contradicts this statement.

We agree with Reviewer 2 that the highlighted sentences were unclear. We have therefore modified the text previously found at lines 78-79:

'We found that HO executed in weeks only a fraction of the tracking work that BACH did in real time. However, HO always outperformed BACH in both ant detection and identification.'

Into:

'We found that HO executed in weeks only a fraction of the tracking work that BACH did in real time. However, HO always made less mistakes and omissions than BACH in both ant detection and identification.' (lines 84-86)

In addition, previous lines 82-83 were unclear most probably due to a typo (we thank Reviewer 2 for spotting it), which we corrected from:

'While BACH identified ants almost as efficiently as HO, its identification performance did not achieve human-like accuracy levels.'

To:

'While BACH detected ant shapes almost as efficiently as HO, its matrix code-based identification performance did not achieve human-like accuracy levels.' (lines 88-90)

Lines 175-177: Please specify the video resolution, so BACH can be better compared to other real-time capable tracking systems, and so it is possible to approximate the size of the barcode in the videos.

We have now modified the sentence to 'We recorded videos at 10 frames/second for 25 seconds and with a resolution of 4096x2160 pixels, in three different conditions:'.

Line 181: The abbreviation "HO" is defined at multiple places in the manuscript. Defining it once would be sufficient.

We have eliminated the repetitions after the first definition of HO at line 79.

Line 322: I was unable to find Table S1.

The sentence referring to a table S1 was a reminiscence of a previous version of the manuscript. We have now deleted it.

Lines 393-395: A better approach would be to use brighter lights so the camera will use a lower exposure time, which will result in less motion blur. Using a higher frame rate will not help to increase the detection rate.

We agree with the suggestion of Reviewer 2 and, also following advice from Reviewer 1, we have now modified the text at lines 457-467 to include a sentence about increasing light intensity:

'In future work, we aim to increase light intensity to shorten cameras' exposure times, reducing the occurrence of blurry images and therefore maximizing code readability.'

Lines 479-480: It is unclear what the error bars represent.

We have now added 'Mean+sd across all panels.' within the figure legend (Lines 612-614).

References:

Mersch, D. P., Crespi, A. & Keller, L. Tracking individuals shows spatial fidelity is a key regulator of ant social organization. *Science* 340, 1090–3 (2013).

Wild, B., Sixt, L. & Landgraf, T. Automatic localization and decoding of honeybee markers using deep convolutional neural networks. 1–20 (2018).

===PREPARING YOUR MANUSCRIPT===

- one version identifying all the changes that have been made (for instance, in coloured highlight, in bold text, or tracked changes);
- a 'clean' version of the new manuscript that incorporates the changes made, but does not highlight them. This version will be used for typesetting if your manuscript is accepted.

===PREPARING YOUR REVISION IN SCHOLARONE===

-- Ensure that your data access statement meets the requirements at <https://royalsociety.org/journals/authors/author-guidelines/#data>. You should ensure that you cite the dataset in your reference list. If you have deposited data etc in the Dryad repository, please include both the 'For publication' link and 'For review' link at this stage.

Journal Name: Royal Society Open Science

Journal Code: RSOS

Online ISSN: 2054-5703

Journal Admin Email: openscience@royalsociety.org

Journal Editor: Andrew Dunn

Journal Editor Email: openscience@royalsociety.org

MS Reference Number: RSOS-202033

Article Status: SUBMITTED

MS Dryad ID: RSOS-202033

MS Title: Integrating real-time data analysis into automatic tracking of social insect behaviour

MS Authors: Sclocco, Alessio; Ong, Shirlyn; Pyay Aung, Sai Yan; Teseo, Serafino

Contact Author: Serafino Teseo

Contact Author Email: steseo@ntu.edu.sg

Contact Author Address 1: 60 Nanyang Drive

Contact Author Address 2:

Contact Author Address 3:

Contact Author City: Singapore

Contact Author State:

Contact Author Country: Singapore

Contact Author ZIP/Postal Code: 637551

Keywords: real-time data analysis, video tracking, ants, animal behaviour

Abstract: Automatic video tracking has become a standard tool for investigating the social behaviour of insects. The recent integration of computer vision in tracking technologies will likely lead to fully automated behavioural pattern classification within the next few years. However, most current systems rely on offline data analysis and use computationally expensive techniques to track pre-recorded videos. To address this gap, we developed BACH (Behaviour Analysis maCHine), a software that performs video tracking of insect groups in real time. BACH uses object recognition via convolutional neural networks and identifies individually tagged insects via an existing matrix code recognition algorithm. We compared the tracking performances of BACH and a human observer across a series of short videos of ants moving in a 2D arena. We found that, concerning computer vision-based ant detection only, BACH performed only slightly worse than the human observer. Contrarily, individual identification only attained human-comparable levels when ants moved relatively slow, and fell when ants walked relatively fast. This happened because BACH had a relatively low efficiency in detecting matrix codes in blurry images of ants walking at high speeds. BACH needs to undergo hardware and software adjustments to overcome its present limits. Nevertheless, our study emphasizes the possibility of, and the need for, integrating real time data analysis into the study of animal behaviour. This will accelerate data generation, visualization and sharing, opening possibilities for conducting fully remote collaborative experiments.

EndDryadContent

Appendix B

Comments to the Author:

I have some few minor comments and suggestions that are listed below. Specific comments relate to the page and line number of the clean version of the word document that was available for review.

Introduction.

Line 72-81. This paragraph presents results, thus I suggest eliminating it from the introduction.

We have now eliminated the results paragraph in the introduction.

Methods and results.

Check decimals. The number of decimals presented is not consistent along the Ms.

Two decimals appear now in numbers across the manuscript, except for p values.

Check \pm symbol along the Ms.

We have now checked and corrected the \pm symbol throughout the manuscript (one instance at line 108).

Discussion.

Line 414-415. Can you please rephrase this sentence. It is not clear.

We have now rephrased from:

'For example, we could match new unknown ant detections with previously detected entities that have become unidentified, merging their status based on numbers and identities of ants tracked in previous frames.'

To

'For example, we could match new unidentified ant detections with previously detected entities that have become unidentified, and then attribute them an identity based on numbers and identities of ants tracked in previous frames.' (lines 405-409)

Line 430-432. Can you please rephrase this sentence. It is not clear.

We have now rephrased from:

'Similar to BACH, other recent tracking systems for insects integrate convolutional networks and individual tags. However, these differ from BACH in their functioning and scopes. For example, AnTrax extracts ant-containing image portions within pre-recorded videos, linking them across frames and reconstructing trajectories for identifying individuals.'

To:

'Similar to BACH, other recent tracking systems for insects integrate convolutional networks and the identification of individual tags. However, such system differ from BACH in their functioning and scopes. For example, AnTrax¹³ extracts image portions including ants from pre-recorded videos, reconstructing ant trajectories and identifying them via linking them across frames.' (lines 423-426)

Line 492-500. Please incorporate references that support the statements.

We have now added references for support (lines 485-493).

Figure captions.

There is no reference to figure 4 in the text.

We have now added references to Fig 4 according to the Editor's comment (lines 314-322).